# Learning Low-Rank Feature for Thorax Disease Classification

**Yancheng Wang**[1]  **Rajeev Goel**[1]  **Utkarsh Nath**[1]  **Alvin C. Silva**[2]  **Teresa Wu**[1]
**Yingzhen Yang**[1]
[1] School of Computing and Augmented Intelligence, Arizona State University
{ywan1053, rgoel15, unath, teresa.wu, yingzhen.yang}@asu.edu
[2] Mayo Clinic Arizona
silva.alvin@mayo.edu

## Abstract

Deep neural networks, including Convolutional Neural Networks (CNNs) and Visual Transformers (ViT), have achieved stunning success in the medical image domain. We study thorax disease classification in this paper. Effective extraction of features for the disease areas is crucial for disease classification on radiographic images. While various neural architectures and training techniques, such as self-supervised learning with contrastive/restorative learning, have been employed for disease classification on radiographic images, there are no principled methods that can effectively reduce the adverse effect of noise and background or non-disease areas on the radiographic images for disease classification. To address this challenge, we propose a novel Low-Rank Feature Learning (LRFL) method in this paper, which is universally applicable to the training of all neural networks. The LRFL method is both empirically motivated by a Low Frequency Property (LFP) and theoretically motivated by our sharp generalization bound for neural networks with low-rank features. LFP not only widely exists in deep neural networks for generic machine learning but also exists in all the thorax medical datasets studied in this paper. In the empirical study, using a neural network such as a ViT or a CNN pre-trained on unlabeled chest X-rays by Masked Autoencoders (MAE), our novel LRFL method is applied on the pre-trained neural network and demonstrates better classification results in terms of both multi-class area under the receiver operating curve (mAUC) and classification accuracy than the current state-of-the-art. The code of LRFL is available at `https://github.com/Statistical-Deep-Learning/LRFL`.

## 1 Introduction

Following the huge success of deep learning, recent studies have developed deep neural networks (DNNs) for various tasks in medical imaging, such as disease classification and abnormalities detection in anatomy in chest X-rays [1, 2]. Accurate clinical decision-making with DNNs heavily relies on learning informative medical feature representation. Early works adopt convolutional neural networks (CNNs) such as U-Net [3] for representation learning on radiography images. Recently, Visual Transformers (ViTs) [4] are also adopted to learn informative medical representations from radiography images [2], utilizing their capabilities in capturing long-range feature dependencies. Albeit the success of CNNs and ViTs in analyzing radiography images, their accuracy heavily relies on the quality and quantity of data and annotations [5]. However, the collection of large amounts of training data and high-quality annotations in the medical imaging domain are extremely hard [2]. To

---

* Indicates equal contribution.

38th Conference on Neural Information Processing Systems (NeurIPS 2024).

tackle this problem, self-supervised learning (SSL) has been employed as a solution for acquiring representations from unlabeled data. Given the greater availability of unlabeled medical images [6], SSL proves to be an efficient approach for obtaining discriminative representations. SSL employs a range of pretext tasks to acquire transferable representations without manual annotations. Over recent years, numerous variations of self-supervised learning have surfaced using contrastive learning [7] and restorative learning [2].

**Challenges in the Current Literature for Disease Classification.** We study thorax disease classification in this paper. Clinical studies show that the disease areas on radiographic images are subtle and exhibit localized variations. Such conditions are further complicated by the inevitable noise that is ubiquitous in radiographic images, as detailed in Section 2.1. Effective and robust extraction of features for the disease areas is crucial for disease classification on radiographic images. Although various neural architectures, such as CNNs and ViTs, and different training techniques, such as self-supervised learning with contrastive/restorative learning, have been employed for disease classification on radiographic images, there have been no principled methods that can effectively reduce the adverse effect of noise and background, or non-disease areas, for disease classification on radiographic images.

**Our Contributions.** The contributions of this paper are presented as follows. First, in order to address the aforementioned challenge, we propose a novel Low-Rank Feature Learning (LRFL) method in this paper, which is universally applicable to the training of all neural networks with the application for thorax disease classification. Our LRFL method employs low-rank features for disease classification. The usage of low-rank features is empirically motivated by a Low Frequency Property (LFP) illustrated in Figure 1. That is, the low-rank projection of the ground truth training class labels possesses the majority of the information of the training class labels. In fact, LFP widely holds for a broad range of classification problems using deep neural networks, such as [1, 8, 9]. Inspired by LFP, our LRFL method adds the truncated nuclear norm as a low-rank regularization term to the training loss of a neural network so as to perform classification using low-rank features. Because the actual features used for classification are approximately low-rank and the high-rank features are significantly truncated, all the noise and the information about the background or the non-disease areas on radiographic images in the high-rank features are largely discarded and not learned in a neural network. *Importantly and significantly different from existing low-rank learning methods reviewed in Section 2.3, we introduce a novel separable approximation for the TNN, enabling the optimization of the LRFL training loss using standard SGD.* The appropriate feature ranks retained in the LRFL method across various datasets are determined through an efficient cross-validation process, and the optimal ranks are detailed in Table 8. Extensive experimental results demonstrate that our LRFL method renders new record mAUC on three standard thorax disease datasets, NIH-ChestX-ray [10], COVIDx [11], and CheXpert [12], surpassing the current state-of-the-art [2] with the same pre-training setup.

Second, we provide a theoretical analysis showing a sharp generalization bound for the LRFL method, underscoring the substantial benefits of employing low-rank regularization within LRFL. Given these theoretical insights and the versatility of LRFL across various neural networks, we anticipate broader applications of LRFL in the classification of other diseases beyond thoracic ones, potentially enhancing classification tasks across different radiographic imaging contexts. It is worthwhile to mention that the literature has studied low-rank learning using TNN resembling LRFL, as to be reviewed in Section 2.3. Our LRFL method builds upon these foundational principles by incorporating low-rank regularization into the training of neural networks, aiming to improve thorax disease classification by reducing the adverse effects of noise and irrelevant background information. **Different from the conventional low-rank learning methods, our approach introduces a separable approximation to the TNN, facilitating the optimization process and enhancing the generalization ability of the model**. Such improved generalization is evidenced by the improved prediction accuracy of LRFL compared to the current state-of-the-art (SOTA) methods in medical image analysis.

Moreover, we have employed a conditional diffusion model trained on COVIDx and CheXpert datasets to generate synthetic images. These synthetic images are then added to their respective training sets to form the augmented training data on which our LRFL models are trained. This approach has further elevated the state-of-the-art mAUC scores achieved by LRFL on both COVIDx and CheXpert datasets.

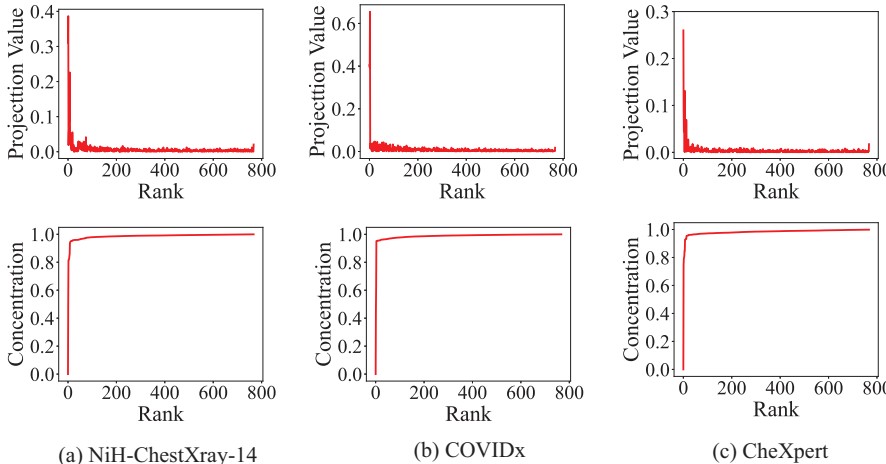

(a) NiH-ChestXray-14    (b) COVIDx    (c) CheXpert

Figure 1: Eigen-projection (first row) and signal concentration ratio (second row) of Vit-Base on NiH-ChestXray-14, COVIDx, and CheXpert. To compute the eigen-projection, we first calculate the eigenvectors $\mathbf{U}$ of the kernel gram matrix $\mathbf{K} \in \mathbb{R}^{n \times n}$ computed by a feature matrix $\mathbf{F} \in \mathbb{R}^{n \times d}$, then the projection value is computed by $\mathbf{p} = \frac{1}{C} \sum_{c=1}^{C} \left\| \mathbf{U}^{\top} \mathbf{Y}^{(c)} \right\|_{2}^{2} / \left\| \mathbf{Y}^{(c)} \right\|_{2}^{2} \in \mathbb{R}^{n}$, where $C$ is the number of classes, and $\mathbf{Y} \in \{0,1\}^{n \times C}$ is the one-hot labels of all the training data, $\mathbf{Y}^{(c)}$ is the $c$-th column of $\mathbf{Y}$. The eigen-projection $\mathbf{p}_r$ for $r \in [\min(n, d)]$ reflects the amount of the signal projected onto the $r$-th eigenvector of $\mathbf{K}$, and the signal concentration ratio of a rank $r$ reflects the proportion of signal projected onto the top $r$ eigenvectors of $\mathbf{K}$. The signal concentration ratio for rank $r$ is computed by $\left\| \mathbf{p}^{(1:r)} \right\|_{2}$, where $\mathbf{p}^{(1:r)}$ contains the first $r$ elements of $\mathbf{p}$. For example, by the rank $r = 38$, the signal concentration ratio of $\mathbf{Y}$ on NIH ChestX-ray14, COVIDx, and CheXpert are 0.959, 0.964, and 0.962 respectively.

**Motivation for using synthetic images to boost the accuracy for thorax disease classification.** The computer vision literature [13, 14, 15] has extensively studied the usage of the generated synthetic images which augment the training data and improve the prediction accuracy of image classification. Inspired and motivated by this observation, we propose to generate synthetic images and use them to form the augmented training data and improve the performance of thorax disease classification. The augmented training data comprise the original training images and the synthetic images. However, too many synthetic images tend to introduce more noise to the augmented training data so excessive synthetic images can hurt the prediction accuracy of DNNs trained on the augmented training data [16]. As evidenced in the ablation study in Section C.3, our proposed LRFL method, coupled with the selection of the number of synthetic images, effectively mitigates this issue. The proposed low-rank learning method only learns the low-rank part of the features learned by a deep learning model so that noise in the high-rank part would not affect the learned model. Also, cross-validation is used to select a proper number of synthetic images, which will boost the prediction accuracy while not introducing too much noise to the augmented training data.

We also present ablation study results evidencing our contributions. We compare the eigenvalues of the kernels and the kernel complexity associated with the LRFL models and the corresponding base models in Section B.4.1 of the appendix, and the lower kernel complexity of the LRFL models suggests their lower generalization error [17, 18, 19].

**Notations.** We use bold letters to denote matrices or vectors. $\left[\mathbf{A}\right]_i$ stands for the $i$-th row of a matrix $\mathbf{A}$. $\|\cdot\|_p$ denotes the $p$-norm of a vector or a matrix. $\|\cdot\|_{\mathrm{F}}$ is the Frobenius norm of a matrix. We use $[m \ldots n]$ to indicate numbers between $m$ and $n$ inclusively, and $[n]$ denotes the natural numbers between 1 and $n$ inclusively.

## 2   Related Works

### 2.1   Radiographic Imaging

Radiographic imaging [20] is a cornerstone in medical image analysis. Unlike photographic images [21], radiography images have consistent backgrounds due to fixed imaging protocols [22, 23, 24, 2]. Clinical details are spread across the images, while areas indicating illness show localized variations [2, 25, 26], making analysis challenging. Noise is unavoidable in ra-

diography images, stemming from quantum fluctuations, electronic interference, scatter radiation, motion blur, and overlapping structures [27, 28, 29, 30]. Quantum noise, originating from statistical fluctuations in detected X-ray photons [31, 32, 25, 30], is often the primary source. Quantum noise introduces graininess, obscuring details and diminishing contrast [31]. Modeled as a Poisson process [25, 30], it can be approximated by a Gaussian distribution under high photon flux [33, 34], enabling noise reduction techniques [34].

## 2.2 Medical Image Analysis with Deep Learning

Deep learning has made remarkable progress in photographic image analysis [35, 36, 37], sparking interest in applying it to medical imaging due to the ability to learn complex representations. Convolutional neural networks (CNNs) like U-Net [3, 38, 39] pioneered this field, achieving state-of-the-art performance across various tasks such as image classification [40, 41, 42], object detection [43, 39, 44], and semantic segmentation [44, 45, 39, 46, 47]. More recently, visual transformers, inspired by the success of transformers in natural language processing [48], have outperformed state-of-the-art CNNs on various computer vision benchmarks [49, 50, 4, 51, 52, 53]. Despite debates around transformers vs CNNs in terms of generalization [54, 55, 56, 57, 58], data requirements [4, 59, 60], and computational costs [61], transformers have shown great potential in medical image analysis [2, 62, 63]. Given the scarcity of high-quality annotations, self-supervised contrastive learning techniques [7, 64, 65, 66, 2] have gained traction for pre-training networks in this domain [22, 2, 62]. However, the high similarity between radiographic images due to standardized protocols [67, 68] poses challenges compared to photographic images [69, 7]. To address this, recent works utilize restorative strategies like masked autoencoders (MAE) [70, 71, 72, 73, 74, 2, 75] for pre-training [2]. Similarly, we adopt MAE [2] to pre-train our networks before learning low-rank features.

## 2.3 Low-Rank Learning

Low-rank learning has garnered significant attention across various fields for its capacity to reduce dimensionality, suppress noise, and enhance feature extraction. Robust Principal Component Analysis (RPCA) [76] serves as a cornerstone in this realm, efficiently separating data matrices into low-rank and sparse components. This technique proves invaluable for vision-related tasks such as image denoising and background subtraction. Building on this foundation, [77] introduced singular value pruning, a method to impose low-rank constraints on neural network layers, thereby boosting both computational efficiency and performance. The concept of TNN regularization (TNNR) has been further refined by researchers like [78], who noted that TNNR more accurately approximates the rank function by selectively minimizing singular values, essential for precise low-rank matrix recovery in noisy conditions. Following that, some existing works [79, 80, 81] propose to perform low-rank feature learning by minimizing the TNN of the feature matrix. Additionally, the use of TNNR in tensor completion has markedly improved the restoration of incomplete visual data, utilizing tensor singular value decomposition (t-SVD) [82, 83]. More contemporary learning-based methods, such as those developed by [84], have optimized low-rank approximations through targeted training, enhancing practical application outcomes. Some works [85, 86, 87] also demonstrate that learning low-rank features can significantly enhance the robustness of deep neural networks against noise in input images. In addition, recent works [88, 89, 90] find that the good generalization capabilities of deep neural networks are attributed to the fact that deeper networks are inductively biased to find solutions with lower effective rank embeddings.

## 3 Formulation

### 3.1 Pipeline for Thorax Disease Classification

We utilize the masked MAE technique [75] for the initial pre-training of both CNNs and ViTs following[2], and subsequently fine-tune the pre-trained networks with our Low-Rank Feature Learning (LRFL). The full training pipeline of learning low-rank features for disease classification can be described in three steps. In the first step, which is the **pre-training** step, we pre-train the networks using the self-supervised restorative learning method, masked MAE [75], on a diverse pre-training dataset that includes ImageNet-1k [91] and a collection of X-rays (0.5M) [2]. During this phase, we randomly mask patches on input images and drive the networks to optimize pixel-wise image reconstruction for the obscured patches. In the second step, which is the **regular fine-tuning** step, we fine-tune the pre-trained networks employing cross-entropy loss aimed at image classification on specific target datasets, namely NIH-ChestX-ray [10], COVIDx [11], and CheXpert [12]. In the

last step, which is the **low-rank feature learning** step, we fix the backbones of the networks and fine-tune the linear classifier utilizing our novel LRFL method.

## 3.2 Problem Setup for LRFL

We now introduce the problem setup for LRFL with training details. Suppose the training data are given as $\{\mathbf{x}_i, \mathbf{y}_i\}_{i=1}^n$ where $\mathbf{x}_i$ and $\mathbf{y}_i \in \mathbb{R}^C$ are the $i$-th training data point and its corresponding class label vector respectively, and $C$ is the number of classes. Each element $\mathbf{y}_i$ is binary with $\mathbf{y}_i = 1$ indicating the $i$-th disease is present in $\mathbf{x}_i$, otherwise $\mathbf{y}_i = 0$. Suppose that the neural network trained by step two of our pipeline in Section 3.1 generates a feature vector $f_{\mathbf{W}_1(0)}(\mathbf{x}) \in \mathbb{R}^d$ (the output of the layer preceding the final linear/softmax layer of the network) for any input x, and $f_{\mathbf{W}'}(\cdot)$ is the feature extraction function with $\mathbf{W}'$ being the weights of the feature extraction backbone of the network. $\mathbf{W}_1(0)$ denotes the denotes the weights of feature extraction backbone by step two of the pipeline. We can train a neural network by optimizing

$$\min_{\mathbf{W}} L(\mathbf{W}) = \frac{1}{n} \sum_{i=1}^n \mathrm{KL}\left(\mathbf{y}_i, \sigma\left(\mathbf{W}_2 f_{\mathbf{W}_1(0)}(\mathbf{x})\right)\right), \tag{1}$$

where $\mathbf{W}_1$ is initialized by $\mathbf{W}_1(0)$, $\mathbf{W}_2 \in \mathbb{R}^{C \times d}$, and $\mathbf{W} = (\mathbf{W}_1, \mathbf{W}_2)$. Here $\sigma$ is an element-wise sigmoid function, $\sigma(\boldsymbol{a}) \in \mathbb{R}^C$ with $[\sigma(\boldsymbol{a})]_c = 1/(1 + \exp(-\boldsymbol{a}_c))$ for $\boldsymbol{a} \in \mathbb{R}^C$ and $c \in [C]$. KL stands for the element-wise binary cross-entropy function. Given two nonnegative vectors $\mathbf{u} = [u_1, \ldots, u_d] \in \mathbb{R}^d, \mathbf{v} = [v_1, \ldots, v_d] \in \mathbb{R}^d$ where $u_i \in \{0, 1\}$ for all $i \in [d]$ and $\|\mathbf{v}\|_\infty \leq 1$, $\mathrm{KL}(\mathbf{u}, \mathbf{v}) := \sum_{j=1}^d -u_i \log v_i - (1 - u_i) \log(1 - v_i)$. We use $\mathbf{Y} = \left[\mathbf{y}_1^\top; \mathbf{y}_2^\top; \ldots; \mathbf{y}_n^\top\right] \in \mathbb{R}^{n \times C}$ to denote the training label matrix by stacking the label vectors of all the training data. Let the mapping function of the neural network used in the loss function $L(\mathbf{W})$ be $\mathrm{NN}_{\mathbf{W}}(\mathbf{x}) = \mathbf{W}_2 f_{\mathbf{W}_1}(\mathbf{x})$.

**Motivation for Low-Rank Regularization** The Low Frequency Property is illustrated in Figure 1, that is, the low-rank projection of the ground truth class labels possesses the majority of the information of the class labels. Inspired by this observation, our LRFL encourages the low-rank part of the feature to participate in the classification process. In this way, the noise and non-disease areas in the high-rank part of the feature are mostly not learned by LRFL so as to improve the classification accuracy. Using notations in Section 3.2, the truncated nuclear norm of $\mathbf{F}$ is $\|\mathbf{F}\|_T := \sum_{i=T+1}^d \sigma_i$ where $T \in [0, d]$. It can be observed by the generalization error bound discussed in Section 3.2 that a smaller $\|\mathbf{F}\|_T$ renders a tighter upper bound for the generalization error of the linear neural network used for LRFL. This observation gives a strong theoretical motivation for us to add the truncated nuclear norm $\|\mathbf{F}\|_T$ to the training loss $L(\mathbf{W})$.

## 3.3 Generalization Bound for Low-Rank Feature Learning

We define the loss function $\ell(\mathrm{NN}_{\mathbf{W}}(\mathbf{x}), \mathbf{y}) := \|\mathrm{NN}_{\mathbf{W}}(\mathbf{x}) - \mathbf{y}\|_2^2$, and the generalization error of the network NN is the expected risk of the loss $\ell$, which is denoted by $L_\mathcal{D}(\mathrm{NN}_{\mathbf{W}}) := \mathbb{E}_{(\mathbf{x}, \mathbf{y}) \sim \mathcal{D}}\left[\ell(\mathrm{NN}_{\mathbf{W}}(\mathbf{x}), \mathbf{y})\right]$, with $\mathcal{D}$ being the distribution of the data x and its class label y. The network $\mathrm{NN}_{\mathbf{W}}$ generates a feature $\mathbf{F} \in \mathbb{R}^{n \times d}$ of all the training data with $\mathbf{F}_i = f_{\mathbf{W}_1}^\top(\mathbf{x}_i)$ for $i \in [n]$. The kernel gram matrix for the feature $\mathbf{F}$ is $\mathbf{K}_n = \frac{1}{n}\mathbf{F}\mathbf{F}^\top$. We let $\widehat{\lambda}_1 \geq \widehat{\lambda}_2 \geq \ldots \geq \widehat{\lambda}_{\bar{r}} > 0$ where $\bar{r} \leq \min\{n, d\}$ is the rank of $\mathbf{K}_n$. Suppose the Singular Value Decomposition of $\mathbf{F}$ is $\mathbf{F} = \mathbf{U}\boldsymbol{\Sigma}\mathbf{V}^\top$, where $\mathbf{U} \in \mathbb{R}^{n \times d}$ has orthogonal columns, $\boldsymbol{\Sigma} \in \mathbb{R}^{d \times d}$ is a diagonal matrix with diagonal elements being the singular values of $\mathbf{F}$, and $\mathbf{V} \in \mathbb{R}^{d \times d}$ is an orthogonal matrix. The columns of $\mathbf{U}$ and $\mathbf{V}$ are also called the left eigenvectors and the right eigenvectors of $\mathbf{F}$, respectively. Let $\sigma_1 \geq \sigma_2 \ldots \geq \sigma_d$ be the singular values of $\mathbf{F}$, and $\bar{\mathbf{Y}} = \mathbf{U}^{(\bar{r})}\mathbf{U}^{(\bar{r})\top}\mathbf{Y}$ be the projection of the training label matrix $\mathbf{Y}$ onto the subspace spanned by the top-$\bar{r}$ left eigenvectors of $\mathbf{F}$, where $\mathbf{U}^{(\bar{r})} \in \mathbb{R}^{n \times \bar{r}}$ is formed by the top $\bar{r}$ eigenvectors in $\mathbf{U}$. Then, we have the following theorem giving the sharp generalization error bound for the linear neural network in (1).

**Theorem 3.1.** For every $x > 0$, with probability at least $1 - \exp(-x)$, after the $t$-th iteration of gradient descent for all $t \geq 1$, we have

$$L_\mathcal{D}(\mathrm{NN}_{\mathbf{W}}) \leq \left\|\mathbf{Y} - \bar{\mathbf{Y}}\right\|_{\mathrm{F}} + c_1\left(1 - \eta\widehat{\lambda}_r\right)^{2t}\|\mathbf{Y}\|_{\mathrm{F}}^2 \quad + c_2 \min_{h \in [0, r]}\left(\frac{h}{n} + \sqrt{\frac{1}{n}\sum_{i=h+1}^r \widehat{\lambda}_i}\right) + \frac{c_3 x}{n}, \tag{2}$$

where $c_1, c_2, c_3$ are positive constants.

**Remark 3.2.** The RHS of (2) is the generalization error bound for the linear neural network used in LRFL as step three of the pipeline in Section 3.1. Moreover, let $\sigma_1 \geq \sigma_2 \ldots \geq \sigma_d$ be the singular values of $\mathbf{F}$. Due to the fact that $\sqrt{\frac{1}{n} \sum_{i=h+1}^{r} \widehat{\lambda}_i} \leq \frac{1}{n} \sum_{i=h+1}^{r} \sigma_i$, it follows by (2) that

$$L_{\mathcal{D}}(\mathrm{NN}_{\mathbf{W}}) \leq c_1 \left(1 - \eta \widehat{\lambda}_r\right)^{2t} \|\mathbf{Y}\|_{\mathrm{F}}^2 + c_2 \left(\frac{h}{n} + \frac{1}{n} \sum_{i=T+1}^{d} \sigma_i\right) + \frac{c_3 x}{n}, \tag{3}$$

which holds for all $T \in [0, d]$. (3) motivates the reduction of the truncated nuclear norm of the feature $\mathbf{F}$, as detailed in the next subsection.

### 3.4 Optimization of the Truncated Nuclear Norm in SGD

The truncated nuclear norm $\|\mathbf{F}\|_T$ is not separable, so the training loss with $\|\mathbf{F}\|_T$ cannot be directly optimized by the standard SGD. To address this problem, we propose an approximation $\overline{\|\mathbf{F}\|_T}$ to $\|\mathbf{F}\|_T$ which is separable so that $\overline{\|\mathbf{F}\|_T}$ can be optimized by standard SGD.

First, we note that if $\mathbf{U}, \mathbf{V}$ are known, then $\mathbf{\Sigma} = \mathbf{U}^\top \mathbf{F} \mathbf{V}$. If we have an approximation $\overline{\mathbf{U}}$ to $\mathbf{U}$ and an approximation $\overline{\mathbf{V}}$ to $\mathbf{V}$, then $\mathbf{\Sigma}$ can be approximated by $\overline{\mathbf{\Sigma}} = \overline{\mathbf{U}}^\top \mathbf{F} \overline{\mathbf{V}}$. As a result, the approximation $\overline{\|\mathbf{F}\|_T}$ to the truncated nuclear norm is $\overline{\|\mathbf{F}\|_T} = \sum_{i=1}^{n} \left( \sum_{s=T+1}^{d} \sum_{k=1}^{d} \overline{\mathbf{U}}_{si}^\top \mathbf{F}_{ik} \overline{\mathbf{V}}_{ks} \right)$. Due to the above discussions, the loss function of LRFL with the approximate truncated nuclear norm $\overline{\|\mathbf{F}\|_T}$ is $\mathcal{L}_{\mathrm{LRFL}}(\mathbf{W}) = \frac{1}{m} \sum_{v_i \in \mathcal{V}_c} \mathrm{KL}\left(\mathbf{y}_i, \left[\sigma\left(\mathbf{F}\mathbf{W}^{(\mathrm{lin})}\right)\right]_i\right) + \eta \overline{\|\mathbf{F}\|_T}$, which is separable, so that it can be trained by the standard SGD. $\eta > 0$ is the weighting parameter for the truncated nuclear norm. Because $\mathcal{L}_{\mathrm{LRFL}}(\mathbf{W})$ is to be optimized by the standard SGD, we have the loss function of LRFL for the $j$-th minibatch $\mathcal{B}_j \subseteq [n]$ as

$$\mathcal{L}_j(\mathbf{W}) = \frac{1}{|\mathcal{B}_j|} \sum_{i \in \mathcal{B}_j} \mathrm{KL}\left(\mathbf{y}_i, \left[\sigma\left(\mathbf{F}\mathbf{W}^{(\mathrm{lin})}\right)\right]_i\right) + \frac{\eta}{|\mathcal{B}_j|} \sum_{i \in \mathcal{B}_j} \left( \sum_{s=T+1}^{d} \sum_{k=1}^{d} \overline{\mathbf{U}}_{si}^\top \mathbf{F}_{ik} \overline{\mathbf{V}}_{ks} \right). \tag{4}$$

The approximation $\overline{\mathbf{U}}$ and $\overline{\mathbf{V}}$ can be computed as the left and right eigenvectors of the feature $\mathbf{F}$ computed at earlier epochs. In order to save computation and avoiding performing SVD for $\mathbf{F}$ at every epoch, we propose to update $\overline{\mathbf{U}}$ and $\overline{\mathbf{V}}$ only after certain epochs. Algorithm 1 describes the training algorithm for the neural network trained with LRFL, which uses the standard SGD to optimize the loss function $\mathcal{L}_{\mathrm{LRFL}}(\mathbf{W})$, as step three of our pipeline in Section 3.1. Before the first epoch, we compute $\overline{\mathbf{U}}$ and $\overline{\mathbf{V}}$ as the left and right eigenvectors of the feature $\mathbf{F}$ at the initialization of the neural network. After every $t_0$ epoch with $t_0$ being a constant integer, we update $\overline{\mathbf{U}}$ and $\overline{\mathbf{V}}$ as the left and right eigenvectors of the feature $\mathbf{F}$ produced by the neural network right after $t_0$-th epoch, with $t_0$ being a constant integer.

---

**Algorithm 1** Training Algorithm with the Approximate Truncated Nuclear Norm by SGD

---

1: Initialize the weights $\mathbf{W}_1$ by $\mathbf{W}_1 = \mathbf{W}_1(0)$, and initialize $\mathbf{W}_2$ randomly
2: Compute feature $\mathbf{F}$ by the neural network, and its SVD as $\mathbf{F} = \mathbf{U}\mathbf{\Sigma}\mathbf{V}$
3: Update $\overline{\mathbf{U}} = \mathbf{U}, \overline{\mathbf{V}} = \mathbf{V}$
4: **for** $t = 1, 2, \ldots, t_{\max}$ **do**
5:     **if** $t \equiv 0 \pmod{t_0}$ **then**
6:         Compute feature $\mathbf{F}$ of the neural network, and its SVD $\mathbf{F} = \mathbf{U}\mathbf{\Sigma}\mathbf{V}$.
7:         Update $\overline{\mathbf{U}} = \mathbf{U}, \overline{\mathbf{V}} = \mathbf{V}$
8:     **end if**
9:     **for** $b = 1, 2, \ldots, B$ **do**
10:         Update $\mathbf{W}$ by applying gradient descent on batch $\mathcal{B}_j \subseteq [n]$ using the gradient of the loss $\mathcal{L}_j$ in Eq.(4)
11:     **end for**
12: **end for**
13: **return** The trained weights $\mathbf{W}$ of the network

---

## 4 Experimental Results

In this section, we conduct experiments on medical datasets to demonstrate the effectiveness of the proposed LRFL method. The experiments section is organized as follows: In Section 4.1, we

discuss our experimental setup and implementation details. In Sections 4.2 and 4.3, we evaluate the LRFL models for thorax disease classification on CheXpert and COVIDx. Evaluation results on NIH ChestX-ray14 are deferred to Section B.1 of the appendix. In Section 4.4, we evaluate synthetic data augmentation on LRFL models, with additional details and results deferred to Section C of the appendix. Comprehensive ablation studies on LRFL are performed in Section 4.5. In Section 4.5.1, we study the effectiveness of the LRFL models in reducing the adverse effect of the background for disease classification. In Section 4.5.2, we study the performance of LRFL models for disease localization. Grad-CAM visualization results of LRFL models and baseline models are illustrated in Section 4.5.3. Additional ablation studies are deferred to Section B.4 of the appendix. In Section B.4.1, we compare the kernel eigenvalues and kernel complexity between the LRFL models and their corresponding base models to show that LRFL improves the generalization capability of the base models by reducing their kernel complexity. In Section B.4.2, we evaluate the performance of the LRFL models with limited data availability. In Section B.4.3, we compare the performance of the LRFL with other fine-tuning strategies. In Section B.4.5, we present the training time of the LRFL models compared with the corresponding base models.

## 4.1 Implementation Details

In this section, we evaluate the proposed LRFL for thorax disease classification. We utilize networks pre-trained on ImageNet [92] or chest X-rays in [2] with MAE, a self-supervised learning strategy that reconstructs missing pixels from patches of input images. We fine-tune these pre-trained networks with low-rank regularization on three public X-ray datasets: (1) NIH ChestX-ray14 [10], (2) Stanford CheXpert [12], and (3) COVIDx [11]. The ADAM optimizer is used with a batch size of 1024 for all datasets. Initially, we fine-tune the entire networks for 75 epochs following the settings in [2], then fine-tune with low-rank regularization for another 75 epochs. We use a cosine learning rate schedule, and the initial learning rate, which is denoted as $\mu$, is selected by cross-validation for each model and each dataset. The default values for momentum and weight decay are set to 0.9 and 0, respectively. We use standard data augmentation techniques, including random-resize cropping, random rotation, and random horizontal flipping. For a fair comparison, all baselines are also fine-tuned for an additional 150 epochs, showing almost no improvement. An exhaustive analysis of this additional fine-tuning is in Section B.4.3. We evaluate our LRFL method on both CNN and visual transformer architectures, including ResNet-50, DenseNet, ViT-S, and ViT-B. Our model is referred to as 'X-LR', where X is the base model (e.g., ResNet-50-LR for ResNet-50 with low-rank features).

**Tuning the $T$, $\eta$, and $\mu$ by Cross-Validation.** We search for the optimal values of feature rank $T$, the weighting parameter for the truncated nuclear norm $\eta$, and the learning rate $\mu$ on each dataset. Let $T = \lceil \gamma \min(n, d) \rceil$, where $\gamma$ is the rank ratio. We select the values of $\gamma$ and $\eta$ by performing 5-fold cross-validation on 20% of the training data in each dataset. The value of $\gamma$ is selected from $\{0.01, 0.02, 0.03, 0.04, 0.05, 0.1, 0.15, 0.2\}$. The value of $\eta$ is selected from $\{5 \times 10^{-4}, 1 \times 10^{-3}, 2.5 \times 10^{-3}, 5 \times 10^{-3}, 1 \times 10^{-2}\}$. The value of $\mu$ is selected form $\{5 \times 10^{-4}, 2.5 \times 10^{-4}, 1 \times 10^{-4}, 5 \times 10^{-5}, 2.5 \times 10^{-5}, 1 \times 10^{-5}\}$. To determine the optimal values of the parameters $\eta$, $\gamma$, and $\mu$, we employ a sequential greedy search strategy. We first fix $\eta$ and $\mu$ and find the optimal value of $\gamma$ by cross-validation. Subsequently, using this optimized $\gamma$, we proceed to search for the optimal $\eta$ while keeping $\mu$ constant. Finally, with optimal $\gamma$ and $\eta$, we search for the optimal $\mu$ by cross-validation. The optimal values of $\eta$, $\gamma$, and $\mu$ selected by cross-validation are shown in Table 8 in Section B.3 of the appendix. The time spent for the entire cross-validation process is presented in Table 9 Section B.3 of the appendix, which demonstrates that the cross-validation process is efficient and does not significantly increase the computational overhead of the training process.

## 4.2 Stanford CheXpert

**Experimental setup.** CheXpert [12] consists of 224,316 chest X-rays collected from 65,240 patients, where 191,028 chest X-rays are used for training. Each X-ray in the dataset has radiology reports indicating the presence of 14 diseases. Following the protocol in [2], all images are resized into $224 \times 224$. We also report the mean AUC (Area Under the Curve) for the 5 distinct classes and conduct a comprehensive comparison with state-of-the-art baseline methods.

**Results and analysis.** Table 1 presents the performance comparisons between the baseline models and the LRFL models on the CheXpert dataset. Throughout this section, we use the postfix "-LR" to

indicate a neural network trained with our LRFL. For example, we use the ViT-B model pre-trained on $489,090$ and the ViT-S model pre-trained on $266,340$ chest X-rays with Masked Autoencoders (MAE) [2]. The pre-trained ViT-B network is fine-tuned on the CheXpert dataset and achieves a mean AUC of 89.3. It is observed that ViT-B-LR achieves state-of-the-art performance of 89.8% in mAUC and improves the performance of ViT-B by 0.5% in mAUC. ViT-S-LR also improves the performance of ViT-S by 0.4% in mAUC, which demonstrates the power of LRFL. We also show the classification accuracy of the five diseases in Table 1, where our method exhibits much better performance than baseline methods. For example, ViT-S-LR achieves an mAUC of 86.3% on Cardiomegaly, with a 4.5% improvement over ViT-S trained with MAE. Such improvements demonstrate the power of LRFL in detecting distinct diseases. The comparison between LRFL models and a more comprehensive list of baseline models are deferred to Table 7 of the appendix.

Table 1: Performance comparisons between LRFL models and SOTA baselines on CheXpert. The best result is highlighted in bold, and the second-best result is underlined. This convention is followed by all the tables in this paper. DN represents DenseNet.

| Method | Architecture | Rank | Atelectasis | Cardiomegaly | Consolidation | Edema | Effusion | mAUC (%) |
|---|---|---|---|---|---|---|---|---|
| Irvin et al.[12] | - | - | 81.8 | 82.8 | _93.8_ | 93.4 | 92.8 | 88.9 |
| Pham et al.[9] | DN121 | - | 82.5 | 85.5 | 93.7 | 93.0 | 92.3 | 89.4 |
| Kang et al.[93] | DN121 | - | 82.1 | 85.9 | **94.4** | 89.2 | 93.6 | 89.0 |
| MoCo v2 [2] | DN121 | - | 78.5 | 77.9 | 92.5 | 92.8 | 92.7 | 88.7 |
| ViT-S [2] | ViT-S/16 | - | _83.5_ | 81.8 | 93.5 | _94.0_ | 93.2 | 89.2 |
| ViT-S-LR (Ours) | ViT-S/16 | 0.05r | **83.7** | _86.3_ | 90.9 | 93.7 | 93.1 | _89.6_ |
| ViT-B [2] | ViT-B/16 | - | 82.7 | 83.5 | 92.5 | 93.8 | **94.1** | 89.3 |
| ViT-B-LR (Ours) | ViT-B/16 | 0.05r | 81.6 | 85.4 | 93.4 | **94.6** | _93.9_ | **89.8** |

## 4.3 COVIDx

**Experimental setup.** COVIDx (Version 9A) [11] consists of 30,386 chest X-rays collected from 17,026 unique patients. We follow the previous works [11, 2] in splitting the dataset into 29,986 training images with four different classes and 400 testing images with three classes. For fair comparisons with the previous methods, we report Top-1 accuracy on the test set (3 classes).

**Results and Analysis.** Table 2 compares the performance of SOTA transformer-based models and the LRFL models on the COVIDx dataset. Similar to Section 4.2, the base ViTs are first pre-trained on chest X-rays using Masked Autoencoders (MAE), and then the pre-trained model is fine-tuned on the COVIDx dataset. It can be observed from Table 2 that both ViT-S-LR and ViT-B-LR outperform their corresponding base models ViT-S and ViT-B, achieving an increase in accuracy of 1.6% and 1.7%, respectively. Table 2 also compares the performance of our LRFL models against the state-of-the-art models on the COVIDx dataset. LRFL models achieve much higher accuracy compared to CNN-based models such as DenseNet-121. ViT-B-LR achieves the new SOTA performance of 97% top-1 accuracy with input resolution set to $224 \times 224$, which exceeds the previous SOTA performance [2] by 1.7% in top-1 accuracy.

Table 2: Performance comparisons between LRFL models and SOTA baselines on COVIDx (in accuracy). DN represents DenseNet.

| Method | Architecture | Rank | Covid-19 Sensitivity | Accuracy |
|---|---|---|---|---|
| COVIDNet-CXR Small [8] | - | - | 87.1 | 92.6 |
| COVIDNet-CXR Large [8] | - | - | 96.8 | 94.4 |
| MoCo v2 [2] | DN121 | - | 94.5 | 94.0 |
| DN121 [2] | DN121 | - | 97.0 | 93.5 |
| ViT-S [2] | ViT-S/16 | - | 94.5 | 95.2 |
| ViT-S-LR (Ours) | ViT-S/16 | 0.01r | _97.5_ | _96.8_ |
| ViT-B [2] | ViT-B/16 | - | 95.5 | 95.3 |
| ViT-B-LR (Ours) | ViT-B/16 | 0.003r | **98.5** | **97.0** |

## 4.4 Improved Results using Diffusion Model

**Experimental Setup.** In this section, we aim to further improve the performance of LRFL models by adding labeled synthetic radiographic images of thorax diseases to the training sets of COVIDx and CheXpert. The synthetic radiographic images are generated by a conditional diffusion model, Diffusion Transformer (DiT) [94], trained on the training set of the corresponding dataset. Details on the training of DiT are deferred to Section C.2 of the appendix. To maintain the same disease co-occurrence, synthetic radiographic images are generated based on the labels from the label set of

each dataset. The number of synthetic images added to the training set of each dataset is determined via cross-validation. We first generate synthetic images of the same size as the training set. The optimal percentage of synthetic images is selected using 5-fold cross-validation on the training data, which is detailed in Section C.2 of the appendix. Synthetic images are combined with the original dataset for further fine-tuning with low-rank regularization. Ablation studies on the number of synthetic images incorporated are performed in SectionC.3.

**Results.** The results of LRFL models trained after adding synthetic images on CheXpert and COVIDx are shown in Table 3. It is observed from the results that adding synthetic data into the training set of LRFL models can further increase their performance. For example, ViT-B-LR with synthetic images added in training outperforms the corresponding base model ViT-B by $2.2\%$ on COVIDx.

Table 3: Performance comparison of baseline models and LRFL models on the CheXpert and COVIDx datasets, with and without synthetic data. $n$ denotes the number of training images in the respective dataset.

| Method | Architecture | CheXpert | | | COVIDx | | |
|---|---|---|---|---|---|---|---|
| | | Rank | # Synthetic Images | mAUC (%) | Rank | # Synthetic Images | Accuracy (%) |
| ViT-S [2] | ViT-S/16 | - | - | 89.2 | - | - | 95.2 |
| ViT-S-LR (Ours) | ViT-S/16 | 0.05r | - | 89.6 | 0.01r | - | 96.8 |
| ViT-S (Ours) | ViT-S/16 | - | $0.2n$ | 89.3 | - | $1.0n$ | 97.0 |
| ViT-S-LR (Ours) | ViT-S/16 | 0.05r | $0.2n$ | 89.7 | 0.01r | $1.0n$ | 97.3 |
| ViT-B [2] | ViT-B/16 | - | - | 89.3 | - | - | 95.3 |
| ViT-B-LR (Ours) | ViT-B/16 | 0.025r | - | 89.8 | 0.003r | - | 97.0 |
| ViT-B (Ours) | ViT-B/16 | - | $0.25n$ | 89.9 | - | $1.0n$ | 97.0 |
| ViT-B-LR (Ours) | ViT-B/16 | 0.025r | $0.25n$ | **90.4** | 0.003r | $1.0n$ | **97.5** |

## 4.5  Ablation Study

### 4.5.1  Study of LRFL in Reducing the Adverse Effects of Background

To demonstrate that the LRFL models are more robust to the background than the baselines, we perform an ablation study on the LRFL to reduce the adverse effects of the background. In this study, we create a mask for the disease area for each original image, then decompose the original image, which has a bounding box for the disease, into a disease image and a background image. Both the disease image and the background image are of the same size as the original image. The background image has grayscale $0$ in the masked disease area, and the disease image has grayscale $0$ in the non-disease area. We feed the three images, which are the original image, the disease image, and the background image, to an LRFL model and obtain the original features, disease features, and background features for the LRFL model, respectively. We also feed these three images to a baseline model and obtain the original features, disease features, and background features for the baseline model. For each original image, we measure the distance between the disease features and original features using KL-divergence on the softmaxed features for the LRFL model and the baseline model. We then compute the average feature distance for each model, which is the average distance between the disease features and original features over the images with a ground-truth bounding box for the disease in the NIH ChestX-ray 14. The results in Table 4 indicate that the original features are closer to the disease features by the LRFL models compared to the baseline models, evidencing the effectiveness of the LRFL models in reducing the adverse effect of the background area. We also remark that since only the low-rank part of the original features participates in the classification process, the noise and non-disease areas in the high-rank part of the features are mostly not learned by LRFL, and in this manner, LRFL is robust to both noise and background.

Table 4: Average feature distance between original features and disease features of images with a ground-truth bounding box for the disease in the NIH ChestX-ray 14.

| Methods | mAUC (%) | Average Feature Distance |
|---|---|---|
| ViT-S | 82.3 | 0.7030 |
| ViT-S-LR | **82.7** | **0.6352** |
| ViT-B | 83.0 | 0.5642 |
| ViT-B-LR | **83.4** | **0.6628** |

### 4.5.2  Disease Localization

To study which part of the X-ray image is responsible for the model prediction by the LRFL models, we perform the disease localization experiment following the settings in [2]. We first obtain the Grad-

CAM visualization results with the last transformer block of ViT-S. The experiments are performed with all the images with a ground-truth bounding box for disease in ChestX-ray14. The predicted bounding box is generated with the thresholded Grad-CAM heatmap, largest connected component, and box regression. We evaluate the performance of disease localization by Intersection over Union (IoU) between the ground-truth bounding box and the predicated bounding box used for evaluation. The Average Precision (AP) on $25\%$ and $50\%$ IoUs, which are denoted as $AP_{25}$ and $AP_{50}$, for ViT-S and ViT-S-LR are shown in Table 5. It is observed from the results that our LRFL model significantly outperforms the base model in detecting the bounding box of thorax disease. For example, ViT-S-LR outperforms ViT-S by $26.9\%$ in $AP_{25}$ for detecting the bounding box of Mass. In addition, ViT-S-LR outperforms ViT-S by $21.2\%$ in $AP_{25}$ for detecting the bounding box of Effusion.

Table 5: $AP_{25}$ and $AP_{50}$ scores for different diseases using ViT-S and ViT-S-LR models.

| Disease | Size (# of px) | $AP_{25}$ | | $AP_{50}$ | |
|---|---|---|---|---|---|
| | | ViT-S | ViT-S-LR | ViT-S | ViT-S-LR |
| Mass | 756 | 27.0 | **53.9** | **11.1** | 8.0 |
| Atelectasis | 924 | 31.5 | **49.3** | 8.1 | **11.3** |
| Pneumothorax | 1899 | 4.7 | **18.3** | 0.0 | **1.5** |
| Infiltrate | 2754 | 11.4 | **22.7** | 1.3 | **2.1** |
| Effusion | 2925 | 8.8 | **30.0** | 1.0 | **3.1** |
| Pneumonia | 2944 | 27.8 | **44.1** | 9.3 | **12.5** |
| All | 2300 | 18.0 | **28.5** | 4.7 | **5.2** |

### 4.5.3 Grad-CAM Visualization

To study how LRFL improves the performance of base models in disease detection, we use the Grad-CAM [95] to visualize the parts in the input images that are responsible for the predictions of the base models and low-rank models. Robust Grad-CAM [95] visualization results of Low-Rank ViT-Base are illustrated in Figure 2. All Grad-CAM visualization results illustrate that our LRFL models usually focus more on the areas inside the bounding box associated with the labeled disease. In contrast, the base models also focus on the areas outside the bounding box or even areas in the background. Robust Grad-CAM visualization results of Low-Rank ResNet-50 and additional Grad-CAM visualization results of Low-Rank ViT-Base are deferred to Figure 4 and Figure 5 in Section B.4.4 of the appendix.

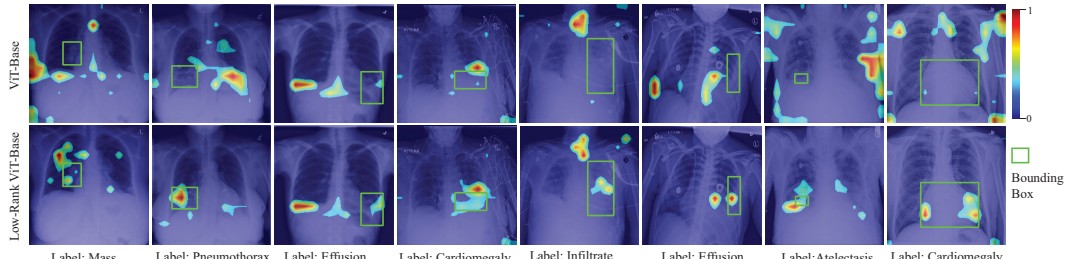

Figure 2: Robust Grad-CAM [95] visualization results on NIH ChestX-ray 14. The figures in the first row are the visualization results of ViT-Base, and the figures in the second row are the visualization results of Low-Rank ViT-Base.

## 5   Conclusion

In this paper, we propose a novel Low-Rank Feature Learning (LRFL) method for thorax disease classification, which can effectively reduce the adverse effect of noise and background, or non-disease areas, on the radiographic images for disease classification. Being universally applicable to the training of all neural networks, LRFL is both empirically motivated by the low frequency property and theoretically motivated by our sharp generalization bound for neural networks with low-rank features. Extensive experimental results on thorax disease datasets, including NIH-ChestX-ray, COVIDx, and CheXpert, demonstrate the superior performance of LRFL in terms of mAUC and classification accuracy. In addition, the performance of LRFL models is further improved by adding synthetic radiographic images into the training set for data augmentation.

## Acknowledgments and Disclosure of Funding

This material is based upon work supported by the U.S. Department of Homeland Security under Grant Award Number 17STQAC00001-07-00. The views and conclusions contained in this document are those of the authors and should not be interpreted as necessarily representing the official policies, either expressed or implied, of the U.S. Department of Homeland Security. This work is also partially supported by the 2023 Mayo Clinic and Arizona State University Alliance for Health Care Collaborative Research Seed Grant Program under Grant Award Number AWD00038846.

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

## A   Proofs

**Proof of Theorem 3.1**. It can be verified that at the $t$-th iteration of gradient descent for $t \geq 1$, we have

$$\mathbf{W}^{(t)} = \mathbf{W}^{(t-1)} - \frac{\eta}{n}\mathbf{F}^\top \left( \mathbf{F}\mathbf{W}^{(t-1)} - \mathbf{Y} \right). \tag{5}$$

It follows by (5) that

$$\mathbf{F}\mathbf{W}^{(t)} = \mathbf{F}\mathbf{W}^{(t-1)} - \eta\mathbf{K}_n\left(\mathbf{F}\mathbf{W}^{(t-1)} - \mathbf{Y}\right)$$
$$= \mathbf{F}\mathbf{W}^{(t-1)} - \eta\mathbf{K}_n\left(\mathbf{F}\mathbf{W}^{(t-1)} - \bar{\mathbf{Y}}\right), \tag{6}$$

where $\mathbf{K}_n = 1/n \cdot \mathbf{F}\mathbf{F}^\top$, $\bar{\mathbf{Y}} = \mathbf{U}^{(\bar{r})}\mathbf{U}^{(\bar{r})^\top}\mathbf{Y}$.

We define $\mathbf{F}(\mathbf{W}, t) := \mathbf{F}\mathbf{W}^{(t)}$, then it follows by (6) that

$$\mathbf{F}(\mathbf{W}, t) - \bar{\mathbf{Y}} = (\mathbf{I}_n - \eta\mathbf{K}_n)\left(\mathbf{F}(\mathbf{W}, t) - \bar{\mathbf{Y}}\right),$$

which indicates that

$$\mathbf{F}(\mathbf{W}, t) - \bar{\mathbf{Y}} = (\mathbf{I}_n - \eta\mathbf{K}_n)^t\left(\mathbf{F}(\mathbf{W}, 0) - \bar{\mathbf{Y}}\right)$$
$$= -(\mathbf{I}_n - \eta\mathbf{K}_n)^t\bar{\mathbf{Y}},$$

and

$$\|\mathbf{F}(\mathbf{W}, t) - \mathbf{Y}\|_{\mathrm{F}} \le \|\mathbf{Y} - \bar{\mathbf{Y}}\|_{\mathrm{F}} + \left(1 - \eta\widehat{\lambda}_r\right)^t\|\bar{\mathbf{Y}}\|_{\mathrm{F}}$$
$$\le \|\mathbf{Y} - \bar{\mathbf{Y}}\|_{\mathrm{F}} + \left(1 - \eta\widehat{\lambda}_r\right)^t\|\mathbf{Y}\|_{\mathrm{F}}. \tag{7}$$

As a result of (7), by using the proof of [17, Theorem 3.3, Corollary 6.7], for every $x > 0$, with probability at least $1 - \exp(-x)$,

$$L_{\mathcal{D}}(\mathrm{NN}_{\mathbf{W}}) \le c_1\|\mathbf{Y} - \bar{\mathbf{Y}}\|_{\mathrm{F}}^2 + c_1\left(1 - \eta\widehat{\lambda}_r\right)^{2t}\|\mathbf{Y}\|_{\mathrm{F}}^2$$
$$+ c_2 \min_{h \in [0,r]}\left(\frac{h}{n} + \sqrt{\frac{1}{n}\sum_{i=h+1}^{r}\widehat{\lambda}_i}\right) + \frac{c_3 x}{n}. \tag{8}$$

$\square$

## B   More Experimental Results

### B.1   NIH ChestX-ray14

**Experimental setup.** NIH ChestX-ray14 [10] consists of $112,120$ X-rays collected from $30,805$ unique patients. Each X-ray can have up to $14$ associated labels, with the possibility of multiple labels per image. Following the official data split in [10], we use $75,312$ images for training and $25,596$ images for testing. The raw images from the dataset are sized $1024 \times 1024$. In our experiments, we scale down the input images to $224 \times 224$. We report the mean AUC (Area Under the Curve) for $14$ distinct classes and conduct a comprehensive comparison with $18$ widely recognized and influential baseline methods.

**Results and Analysis.** Table 6 presents the performance comparisons between several top-performing baseline models and their corresponding low-rank models on the NIH ChestX-ray14 dataset. Similar to Section 4.2, the ViTs are first pre-trained chest X-rays using Masked Autoencoders (MAE). Then, the pre-trained ViT-B network is fine-tuned on the NIH ChestX-ray14 dataset and achieves a mean AUC of $83.0$. Next, we fine-tune ViT-B with low-rank regularization for another 75 epochs. The low-rank model, denoted as ViT-B-LR, achieves the new state-of-the-art performance with a mean AUC of $83.4$. It is observed that all low-rank models achieve improvement in mean AUC compared to the corresponding base models. It is important to highlight that the research community dedicated four years to enhancing the AUC score for CNN-type architectures, advancing it from $74.5\%$ to $82.2\%$, which was primarily attributed to the challenging nature of the classification with the NIH ChestX-ray14 dataset.

### B.2   Comparison with A Comprehensive List of Baselines

We compare the results of LRFL models with a more comprehensive list of baselines on CheXpert. It is observed from the results in Table 7 that LRFL models significantly outperform all existing state-of-the-art methods on CheXpert.

Table 6: Performance comparisons between LRFL models and SOTA baselines on NIH ChestX-ray14. RN, DN, and SwinT represent ResNet, DenseNet, and Swin Transformer.

| Method | Architecture | Pre-training | Rank | mAUC |
|---|---|---|---|---|
| Wang et al. [10] | RN50 | | - | 74.5 |
| Li et al.[96] | RN50 | | - | 75.5 |
| Yao et al. [97] | RN&DN | | - | 76.1 |
| Wang et al.[41] | R152 | | - | 78.8 |
| Ma et al.[98] | R101 | | - | 79.4 |
| Tang et al.[99] | RN50 | | - | 80.3 |
| Baltruschat et al.[100] | RN50 | | - | 80.6 |
| Guendel et al.[1] | DN121 | | - | 80.7 |
| Guan et al.[101] | DN121 | ImageNet-1K | - | 81.6 |
| Seyyed et al.[102] | DN121 | | - | 81.2 |
| Ma et al.[42] | DN121(×2) | | - | 81.7 |
| Hermoza et al.[103] | DN121 | | - | 82.1 |
| Kim et al.[104] | DN121 | | - | 82.2 |
| Haghighi et al.[68] | DN121 | | - | 81.7 |
| Liu et al.[105] | DN121 | | - | 81.8 |
| Taslimi et al.[106] | SwinT | | - | 81.0 |
| MoCo v2 [2] | DN121 | X-rays (0.3M) | - | 80.6 |
| MAE [2] | DN121 | | - | 81.2 |
| RN-50 [2] | RN50 | ImageNet-1K | - | 81.8 |
| RN-50-LR (Ours) | RN50 | | 0.05r | 82.2 |
| DN-121 [2] | DN121 | ImageNet-1K | - | 82.0 |
| DN-121-LR (Ours) | DN121 | | 0.05r | 82.4 |
| ViT-S [2] | ViT-S/16 | X-rays (0.3M) | - | 82.3 |
| ViT-S-LR (Ours) | ViT-S/16 | | 0.05r | 82.7 |
| ViT-B [2] | ViT-B/16 | X-rays (0.5M) | - | 83.0 |
| ViT-B-LR (Ours) | ViT-B/16 | | 0.05r | **83.4** |

Table 7: The table shows the performance of various state-of-the-art (SOTA) CNN-based and Transformer- based methods on CheXpert.

| Method | Architecture | Rank | Atelectasis | Cardiomegaly | Consolidation | Edema | Effusion | mAUC (%) |
|---|---|---|---|---|---|---|---|---|
| Allaouzi et al.[107] | | - | 72.0 | **88.0** | 77.0 | 87.0 | 90.0 | 82.8 |
| Irvin et al.[12] | | - | 81.8 | 82.8 | 93.8 | 93.4 | 92.8 | 88.9 |
| Seyyedkalantari et al.[102] | | - | 81.2 | 83.0 | 90.0 | 88.3 | 93.8 | 87.3 |
| Pham et al.[9] | | - | 82.5 | 85.5 | 93.7 | 93.0 | 92.3 | 89.4 |
| Hosseinzadeh et al.[108] | DN121 | - | - | - | - | - | - | 87.1 |
| Haghighi et al.[68] | | - | - | - | - | - | - | 87.6 |
| Kang et al.[93] | | - | 82.1 | 85.9 | **94.4** | 89.2 | 93.6 | 89.0 |
| DN121 (MoCo v2) [2] | | - | 78.5 | 77.9 | 92.5 | 92.8 | 92.7 | 88.7 |
| DN121 [2] | | - | 81.5 | 77.6 | 89.4 | 92.3 | 92.0 | 88.7 |
| ViT-S [2] | ViT-S/16 | - | 83.5 | 81.8 | 93.5 | 94.0 | 93.2 | 89.2 |
| ViT-S-LR (Ours) | ViT-S/16 | 0.05r | **83.7** | 86.3 | 90.9 | 93.7 | 93.1 | 89.6 |
| ViT-B [2] | ViT-B/16 | - | 82.7 | 83.5 | 92.5 | 93.8 | **94.1** | 89.3 |
| ViT-B-LR (Ours) | ViT-B/16 | 0.05r | 81.6 | 85.4 | 93.4 | **94.6** | 93.9 | **89.8** |

## B.3   Cross-Validation Results

The optimal values of the rank ratio $\gamma$, weighting parameter $\eta$, and learning rate $\mu$ decided by cross-validation for different models on different datasets are shown in Table 8.

Table 8: Optimal values of rank ratio $\gamma$, weighting parameter $\eta$, and learning rate $\mu$ decided by cross-validation for different models on different datasets.

| Models | Parameters | NIH-ChestX-ray | COVIDx | CheXpert |
|---|---|---|---|---|
| ViT-S | $\gamma$ | 0.05 | 0.01 | 0.05 |
| | $\eta$ | $5 \times 10^{-4}$ | $1 \times 10^{-3}$ | $1 \times 10^{-3}$ |
| | $\mu$ | $5 \times 10^{-5}$ | $2.5 \times 10^{-5}$ | $1 \times 10^{-5}$ |
| ViT-B | $\gamma$ | 0.05 | 0.003 | 0.05 |
| | $\eta$ | $5 \times 10^{-4}$ | $1 \times 10^{-3}$ | $1 \times 10^{-3}$ |
| | $\mu$ | $5 \times 10^{-5}$ | $2.5 \times 10^{-5}$ | $2.5 \times 10^{-5}$ |

In addition, the time for the entire cross-validation process in searching for the optimal values of the rank ratio $\gamma$, weighting parameter $\eta$, and learning rate $\mu$ are shown in Table 9. The evaluation is performed on 4 Nvidia A100 GPUs. As we use only 20% of the training data for cross-validation and train the models with each option for only 40% of the entire number of training epochs, the entire

cross-validation process is efficient and does not largely increase the computation cost of the training process.

Table 9: Time Spent for cross-validation on NIH ChestX-ray14, CheXpert, and CovidX. All the results are reported in minutes.

| Datasets | NIH ChestX-ray14 | CheXpert | CovidX |
|---|---|---|---|
| ViT-S-LR | 149 | 178 | 57 |
| ViT-B-LR | 172 | 285 | 69 |

## B.4 Additional Ablation Study

### B.4.1 Study on the Kernel Eigenvalues and Kernel Complexity

Kernel complexity [17, 18, 19] is a widely-studied complexity measure for the generalization capability of kernel-based learning algorithms. In this section, we compare the eigenvalues of the kernel and kernel complexity of ViT-B-LR and ViT-B on ChestX-ray14, COVIDx, and CheXpert. Given the representations of all the training images $\mathbf{F}$ learned by ViT-B or ViT-B-LR, the kernel complexity of the gram matrix $\mathbf{K}_n = \frac{1}{n}\mathbf{FF}^\top$, which is also defined in Section 3.3, can be computed by $\min_{h \in [0,n]} \left( \frac{h}{n} + \sqrt{\frac{\sum_{i=h+1}^{n} \widehat{\lambda}_i}{n}} \right)$.

The eigenvalues of ViT-B-LR and ViT-B on ChestX-ray14, COVIDx, and CheXpert are illustrated in Figure 3. The computed kernel complexities of ViT-B-LR and ViT-B on ChestX-ray14, COVIDx, and CheXpert are shown in Table 10. It is observed that LRFL significantly reduces the kernel complexity of the image representations, which suggests that the LRFL models have lower generalization errors [17, 18, 19].

Table 10: Kernel complexity comparison between ViT-B-LR and ViT-B on ChestX-ray14, COVIDx, and CheXpert.

| Method | ChestX-ray14 | | COVIDx | | CheXpert | |
|---|---|---|---|---|---|---|
| | Kernel Complexity | h | Kernel Complexity | h | Kernel Complexity | h |
| ViT-B | 0.0101 | 465 | 0.0207 | 303 | 0.0040 | 766 |
| ViT-B-LR | 0.0076 | 262 | 0.0155 | 187 | 0.0038 | 389 |

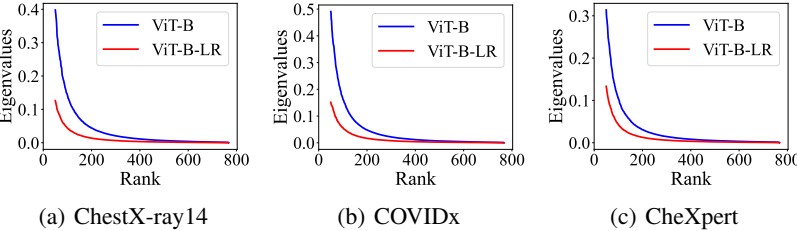

(a) ChestX-ray14      (b) COVIDx      (c) CheXpert

Figure 3: Eigenvalues comparison between ViT-B-LR and ViT-B on ChestX-ray14, COVIDx, and CheXpert.

### B.4.2 Experiments in Small Data Regimes

**Experimental setup.** We explore the effectiveness of low-rank features learned in scenarios with limited data availability, which is particularly significant given the challenges in acquiring high-quality data annotations in the medical imaging domain. We expect that LRFL models can demonstrate improved performance in such situations due to our theoretical guarantee of the better generalization capability of LRFL. We randomly select 5%, 10%, 15%, 20%, 25%, and 50% of the training data from the NIH ChestX-ray14 dataset and then fine-tune the base model using its default training configurations. We then train LRFL models for 20 epochs.

**Results and analysis.** As depicted in Table 11, our LRFL models consistently outperform their corresponding base methods across all data subsets, including 5%, 10%, 15%, 20%, 25%, and 50% on the NIH ChestX-ray14 dataset. Notably, the average improvement in performance is more substantial for the 5% data subset compared to the remaining subsets. For instance, ViT-B-LR exhibits a remarkable improvement of 1.05% for the 5% data subset, which significantly surpasses the improvements of 0.15%, 0.06%, 0.06%, 0.09%, and 0.11% observed for the 10%, 15%, 20%, 25%, and 50% training data subsets, respectively. These findings are consistent with our expectations, showcasing the strong generalization capability of LRFL models in mitigating over-fitting issues with limited data. In conclusion, our findings in the low-data regimes demonstrate the superiority of our LRFL in delivering more generalizable and robust representations for tasks with limited data availability, thereby contributing to the reduction of annotation costs.

Table 11: The table evaluates the performance of various models under low data regimes on the NIH ChestX-rays14 dataset. Models trained with low-rank features effectively combat overfitting in scenarios with limited data availability, thereby enhancing the quality of representations for downstream tasks.

| Pre-training Dataset | Model | Label Fractions | | | | | | | | | | | |
|---|---|---|---|---|---|---|---|---|---|---|---|---|---|
| | | 5% | | 10% | | 15% | | 20% | | 25% | | 50% | |
| | | Rank | mAUC | Rank | mAUC | Rank | mAUC | Rank | mAUC | Rank | mAUC | Rank | mAUC |
| X-rays(0.3M) | ViT-S | - | 61.22 | - | 73.19 | - | 76.99 | - | 78.65 | - | 79.57 | - | 81.20 |
| | ViT-S-LR(Ours) | $0.05r$ | 61.81 | $0.2r$ | 73.84 | $0.04r$ | 77.21 | $0.04r$ | 78.86 | $0.05r$ | 79.65 | $0.05r$ | 81.35 |
| X-rays(0.5M) | ViT-B | - | 70.71 | - | 78.67 | - | 79.99 | - | 80.59 | - | 81.13 | - | 82.19 |
| | ViT-B-LR (Ours) | $0.05r$ | 71.76 | $0.2r$ | 78.82 | $0.2r$ | 80.05 | $0.1r$ | 80.65 | $0.05r$ | 81.22 | $0.05r$ | 82.30 |

### B.4.3    Exploring Fine-tuning Strategies

Our LRFL method learns low-rank features by leveraging models pre-trained on the target dataset. In this section, we conduct an ablation study to investigate the significance of low-rank regularization in the fine-tuning process. A detailed comparative analysis of low-rank regularization against several performance-enhancing techniques, including mix-up [109], label smoothing [110], and EMA [111], is presented in Table 12. We performed an experiment by fine-tuning without low-rank regularization and other tricks, which serves as a baseline for studying the effects of fine-tuning strategies. All models underwent equivalent training epochs to ensure a fair comparison. The results demonstrate that LRFL models achieve the highest performance improvement compared to all other approaches. Notably, unlike natural images, applying mix-up, label smoothing, or EMA to the NIH ChestX-ray dataset leads to performance drops (see Table 12). Fine-tuning models pre-trained on the target dataset without low-rank regularization does not lead to performance improvements compared to fine-tuning with low-rank regularization. For example, the original ViT-S [2] achieves a mean AUC of 82.27% on NIH Chest Xray-14. Fine-tuning this model for 20 epochs without low-rank regularization leads to a mean AUC of 82.26%, whereas fine-tuning with low-rank regularization for 75 epochs results in a mean AUC of 83.40%. We observe similar results for all models based on low-rank features, demonstrating the significance of LRFL.

Table 12: Comparison of fine-tuning strategies on NIH ChestX-ray14.

| Model | mAUC | | | | | |
|---|---|---|---|---|---|---|
| | Base Model | Fine-tuning | Mix-up [109] | Label Smoothing [110] | EMA [111] | LRFL |
| ViT-S | 82.27 | 82.26 | 82.09 | 82.24 | 82.26 | 82.70 |
| ViT-B | 83.00 | 83.00 | 82.37 | 82.99 | 82.98 | **83.40** |

### B.4.4    Additional Grad-CAM Visualization Results

Additional Grad-CAM visualization results of the Low-Rank ViT-Base on NIH ChestX-ray 14 are illustrated in Figure 5. Robust Grad-CAM visualization results of the Low-Rank ResNet-50 are illustrated in Figure 4. We visualize the parts in the input images that are responsible for the predictions of the ground-truth disease label for base models and low-rank models. The visualization results show that our low-rank models usually focus more on the areas inside the bounding box associated with the labeled disease. In contrast, the base models also focus on the areas outside the bounding box or even areas in the background.

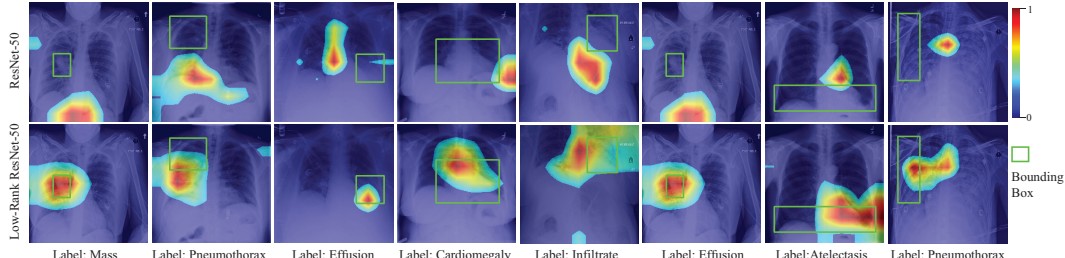

Figure 4: Robust Grad-CAM [95] visualization results on NIH ChestX-ray 14. The figures in the first row are the visualization results of ViT-Base, and the figures in the second row are the visualization results of Low-Rank ResNet-50.

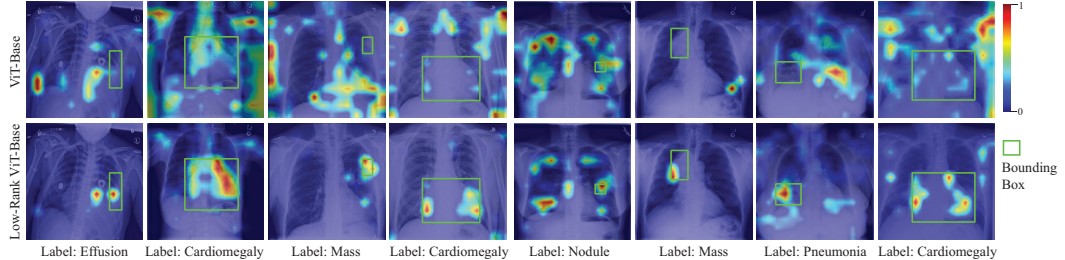

Figure 5: Grad-CAM visualization results on NIH ChestX-ray 14. The figures in the first row are the visualization results of ViT-Base, and the figures in the second row are the visualization results of Low-Rank ViT-Base.

### B.4.5   Training Time Analysis

We evaluate the training time of LRFL models and compare them with the training time of the baseline models. The evaluation of LRFL models and baseline models is performed on 4 Nvidia A100 GPUs. It is observed from the results in Table 13 that the training time of LRFL models is comparable to the training time of LRFL models. The main computational overhead of LRFL models is the computation of the eigenvectors of the feature matrix $\mathbf{F}$ and the truncated nuclear norm. However, the computation overhead is largely reduced by avoiding performing SVD for the feature matrix $\mathbf{F}$ at every epoch, benefiting from the approximation algorithm we designed in Algorithm 1.

Table 13: Training time comparison between LRFL models and baseline models on NIH ChestX-ray14, CheXpert, and CovidX. All the results are reported in minutes.

| Datasets | NIH ChestX-ray14 | CheXpert | CovidX |
|---|---|---|---|
| ViT-S | 54 | 90 | 23 |
| ViT-S-LR | 98 | 117 | 38 |
| ViT-B | 72 | 162 | 32 |
| ViT-B-LR | 113 | 185 | 45 |

## C   Training with Synthetic Data by Diffusion Models

In this section, we explore generative data augmentation using diffusion models. Section C.1 introduces the preliminaries of diffusion models and outlines the specific modifications made for our target task. In Section C.2, we discuss the implementation details of training of the diffusion model. Finally, we show some of the generated synthetic images in Figure 6.

### C.1   Data Generation with the Diffusion Model

The diffusion model operates through a probabilistic framework, employing a forward noising process that gradually introduces noise to the original data $\mathbf{x}_0$. Initially, the model defines a distribution

$q(\mathbf{x}_t|\mathbf{x}_0)$ where $\mathbf{x}_t$ is progressively noised from $\mathbf{x}_0$ over time $t$. This distribution is governed by predetermined hyperparameters $\bar{\alpha}_t$, with $\mathbf{x}_t$ sampled using the reparameterization trick $q(\mathbf{x}_t|\mathbf{x}_0) = \mathcal{N}(\mathbf{x}_t; \sqrt{\bar{\alpha}_t}\mathbf{x}_0, (1 - \bar{\alpha}_t)\mathbf{I})$, where $\epsilon \sim \mathcal{N}(0, \mathbf{I})$. As $t$ advances, the noise contribution increases, leading $\mathbf{x}_t$ to become progressively noisier until it approximates a standard Gaussian distribution.

Following the training of the diffusion model, the focus shifts to the reverse process, aimed at denoising a noisy sample $\mathbf{x}_t$ to recover the original data $\mathbf{x}_0$. Utilizing a Gaussian noise $\mathbf{x}_T$ as the starting point, the model iteratively refines the sample using $p_\theta(\mathbf{x}_{t-1}|\mathbf{x}_t)$, where $\mu_\theta$ is parameterized to approximate the posterior mean of the forward process $p_\theta(\mathbf{x}_{t-1}|\mathbf{x}_t) = \mathcal{N}(\mathbf{x}_{t-1}; \mu_\theta(\mathbf{x}_t, t), \Sigma_\theta(\mathbf{x}_t, t))$, where $\mu_\theta(\mathbf{x}_t, t) = \frac{1}{\sqrt{\alpha_t}}\left(\mathbf{x}_t - \frac{1-\alpha_t}{\sqrt{1-\bar{\alpha}_t}}\epsilon_\theta(\mathbf{x}_t, t)\right)$. Training involves minimizing a simplified loss function $L_{\text{simple}}$ to ensure accurate prediction of noise, facilitating effective denoising during the reverse process $L_{\text{simple}} = \mathbb{E}_{t, \mathbf{x}_0, \epsilon}\left[\|\epsilon - \epsilon_\theta(\mathbf{x}_t, t)\|^2\right]$.

We adopt a class of diffusion model known as the Diffusion Transformer (DiT) [94], chosen for its efficiency and token-agnostic conditioning, making it particularly suitable for our task. DiTs efficiently leverage label embedding for guidance and exhibit high compute efficiency, which is crucial for scaling to large datasets.

The Diffusion Transformer (DiT) model is trained on the CheXpert and COVIDx datasets, as described by [94]. The DiT model, specifically designed for text-based labels, operates without classifiers and instead relies on label embedding to guide the diffusion process. Modifications are made to the label embedding layer to adapt the model to the multi-label problem. Once the training is finished, images are sampled according to the label distribution of the original dataset to maintain the distribution of co-occurring diseases in the synthetic dataset.

## C.2 Implementation Details

**Training Settings of the Diffusion Model.** Following the protocol in [94], the DiT is trained on $256 \times 256$ images for $2800$ epochs, employing a global batch size of $512$ distributed across 4 Nvidia A100 GPUs. Throughout training, a constant learning rate of $10^{-4}$ is maintained. After the training of the diffusion model is finished, synthetic images are sampled using a CFG scale of $4.0$ and $128$ sampling steps. To preserve the disease co-occurrence distribution within the synthetic dataset, identical image labels as those from the original dataset are utilized. The number of synthetic images added to the training set of each dataset is determined via cross-validation. We first generate synthetic images of the same size as the training set. The optimal percentage of synthetic images is selected using 5-fold cross-validation on the training data. Synthetic images are combined with the original dataset for further fine-tuning with low-rank regularization. Figure 6 presents examples of the synthetic chest X-rays generated using the aforementioned setting.

Table 14: Selected optimal percentage of images $\alpha$ on different datasets and models.

| Dataset | CheXpert | | | | COVIDx | | | |
|---|---|---|---|---|---|---|---|---|
| Models | ViT-S | ViT-S-LR | ViT-B-LR | ViT-B-LR | ViT-S | ViT-S-LR | ViT-B | ViT-B-LR |
| $\alpha$ | 0.15 | 0.2 | 0.25 | 0.25 | 0.7 | 1.0 | 0.75 | 1.0 |

**Tuning the Number of Synthetic Images $n$ by Cross-Validation.** We determine the optimal number of synthetic images for each dataset and the corresponding ViT variant. Let $N = \lceil \alpha \times n \rceil$, where $\alpha$ is the percentage of the images and $n$ denotes the size of the training set of the target dataset. The values of $\alpha$ are selected through 5-fold cross-validation on the training data in each dataset. Specifically, $\alpha$ is chosen from the set $\{0.1, 0.15, 0.2, 0.25, 0.3, 0.35, 0.4, 0.45, 0.5, 1.0\}$. The optimal values of $\alpha$ selected by cross-validation for each dataset and ViT variant are presented in Table 14.

**Training Settings of the LRFL Models with the Synthetic Images.** Once we obtain the synthetic images generated by the diffusion model, we add the synthetic images into the training set of the target datasets, including COVIDx and CheXpert. We also leverage networks pre-trained on ImageNet [92] or chest X-rays [2] using Masked Autoencoders (MAE). The MAE pre-trained models are fine-tuned following the same pipeline as in Section 3.1 and the same implementation details as in Section 4.1.

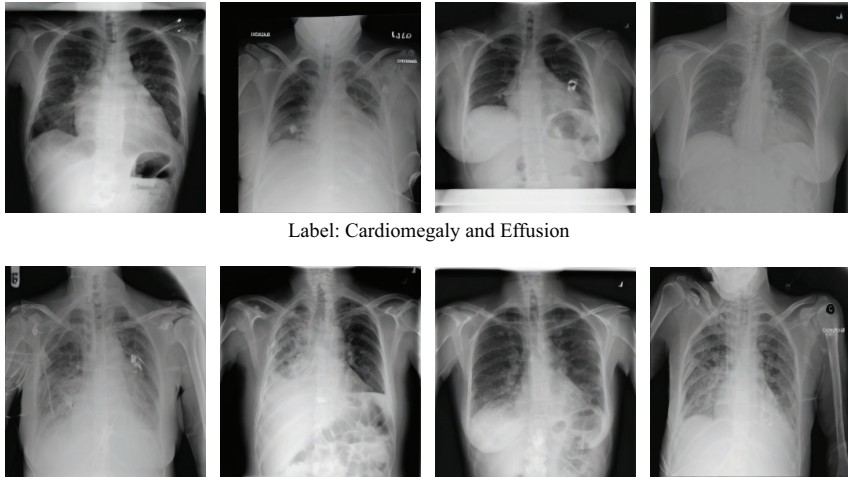

Label: Cardiomegaly and Effusion

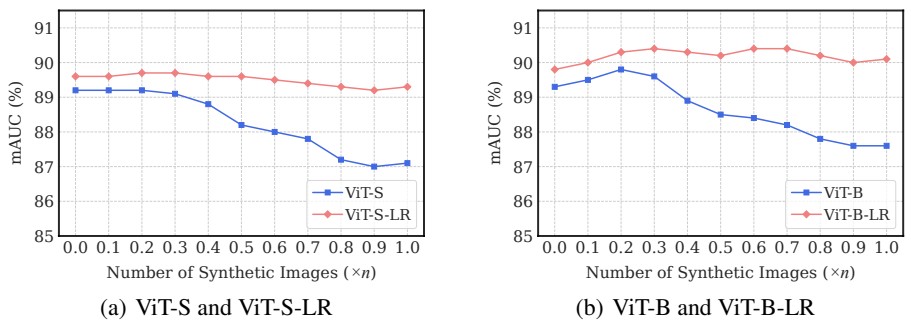

Label: Effusion, Infiltration and Pneumonia

Figure 6: Synthetic images generated using the Diffusion Model. The images in the first row are labeled Cardiomegaly and Effusion, and the images in the second row are labeled Effusion, Infiltration, and Pneumonia.

(a) ViT-S and ViT-S-LR

(b) ViT-B and ViT-B-LR

Figure 7: Performance comparisons between base models and LRFL models trained with different numbers of synthetic images added on CheXpert. $n$ is the number of original training images in CheXpert.

## C.3  Ablation Study on the Number of Synthetic Images

Although the usage of the generated synthetic images can improve the prediction accuracy of DNNs for image classification [13, 14, 15], too many synthetic images tend to introduce more noise to the augmented training data so excessive synthetic images can hurt the prediction accuracy of DNNs trained on the augmented training data [14]. Our proposed LRFL method, coupled with the selection of the amount of synthetic images, effectively mitigates this issue. In this section, we compare the performance of LRFL models with base models when different numbers of synthetic images are added to the training set of CheXpert. As illustrated in Figure 7, the performance of both the LRFL model and the base model can be initially improved with more synthetic images. However, after a certain point, even more, synthetic images start to hurt the performance due to the noise in the synthetic images, and the literature on using synthetic data for training classifiers such as [13] also has a similar observation. This is the reason why we need to perform a cross-validation on the size of the synthetic data for the best performance. Importantly, it can be observed that our LRFL models (ViT-S-LR or ViT-B-LR) usually improve the performance of the corresponding base models (ViT-S or ViT-B) on different choices of the size of the synthetic data. The improvements of our LRFL models over the corresponding base models tend to be more significant as the size of synthetic data increases. This observation justifies the effectiveness of LRFL in reducing the adverse effect of noise in the synthetic images. For example, ViT-B-LR outperforms ViT-B by $0.5\%$ in mAUC when $0.1n$ synthetic images are added into the training set, and the improvement escalates to $2.5\%$ with $n$ synthetic images added into the training set where $n$ is the size of the original training data.

