# OpenReview forum: "Learning Low-Rank Feature for Thorax Disease Classification"
_NeurIPS.cc/2024/Conference — NeurIPS 2024 poster_

### Official Review · Reviewer_JazK · 2024-07-08

**Soundness:** 2
**Presentation:** 2
**Contribution:** 2
**Rating:** 5
**Confidence:** 4

**Summary:**

The authors proposed to use Low-rank Feature Learning (LRFL) to improve model performance specifically for thorax diseases classification. Ideas come from the assumption that the low-rank features capture the majority of the information. They implemented the LRFL by adding a self-modified regularization term, and provided theoretical results on the boundaries of the LRFL methods.

**Strengths:**

Rigorous proof, nice writing and relative comprehensive experimentations.

**Weaknesses:**

It seems like the authors was tackling a very large and general topic on improving model performance by learning low-rank features for some specific. There is no specific design related to thorax disease, the X-ray image, or even the medical image. I would assume this method could be applied to any imaging with some background noise. Therefore, since there are existing studies doing low-rank features-related experiments, I would see this is a similar implementation of the low-rank feature learning but very hard to observe valuable novelties.

**Questions:**

. “truncated nuclear norm as” how did the truncate nuclear norm work? May need references.
	2. “Because the actual features used for classification are approximately low-rank and the high-rank features are significantly truncated, all the noise and the information about the background, or the non-disease areas on radiographic images in the high-rank features are largely discarded and not learned in a neural network.” Any reference to support your statement?
	3. How does the image dimensions (HW dim & C dim) matched when you use both datasets “ImageNet-1k and X-rays (0.5M)” for pretraining?
	4. Table 2, how the decision cutting threshold was chosen? Is it the same across different models/datasets?
	5. I have a different thought against the authors. I understand that the low-rank features can maintain the majority of the information. However, some of the nodules or lesions or abnormalities that used for diagnosis are actually small and tiny. Then, by using the low-rank feature approach, the small region information might be lost but will not negatively impact the over-all information that much. How does the method handle this situation?
	6. It seems like the authors proposed a very large and general topic on improving model performance by learning low-rank features for some specific. There is no specific design related to thorax disease, the X-ray image, or even the medical image. I would assume this method could be applied to any imaging with some background noise. Therefore, since there are existing studies doing low-rank features-related experiments, I would see this is a similar implementation of the low-rank feature learning but very hard to observe valuable novelties.

---

> ### Author Rebuttal · Authors · 2024-08-07
>
> We appreciate the review and the suggestions in this review. The raised issues are addressed below.
>
> **1. ...how did the truncated nuclear norm work? May need references.**
>
> The truncated nuclear norm is defined in line 171 of our paper. Existing works [1,3, 4] perform low-rank learning by minimizing the truncated nuclear norm of the feature matrix.
>
> **2. “Because the actual features used for classification are approximately low-rank and the high-rank features are significantly truncated, all the noise and the information about the background, or the non-disease areas on radiographic images in the high-rank features are largely discarded and not learned in a neural network.” Any reference to support your statement?**
>
> Studies in the literature [5,6,7] show that learning low-rank features can enhance robustness to noise in the input images. We will add these references to our paper to support the claim.
>
> **3. How does the image dimensions (HW dim & C dim) matched when you use both datasets “ImageNet-1k and X-rays (0.5M)” for pretraining?**
>
> When ImageNet-1k and X-rays (0.5M) are used for the pre-training of models in our paper, all the images will be reshaped to $224 \times 224 \times 3$ following the settings in [8].
>
> **4. Table 2, how the decision cutting threshold was chosen?...**
>
> We performed cross-validation described in line 248-261 to decide the cutting threshold for the rank in Table 2. Yes, we performed the same cross-validation process across different models/datasets.
>
> **5. I have a different thought against the authors...**
>
> We respectfully disagree with your thought that “by using the low-rank feature approach, the small region information might be lost”. We now demonstrate the evidence that the features of small disease regions are still captured well by our LRFL model. In particular, because ViT is known to be robust to diseases in a small size such as nodules [2], LRFL on a ViT model can still capture such small-sized disease areas. This is evidenced by the results in Average Precision (AP) about disease localization in point two of our response to Reviewer cPxj or point 3 of our global response. In these results, it is shown that LRFL model renders higher AP for small disease areas such as nodules for disease localization.
>
> **6. ...I would see this is a similar implementation of the low-rank feature learning but very hard to observe valuable novelties.***
>
> We respectfully but strongly disagree with the claim that “I would see this is a similar implementation of the low-rank feature learning”. The significant contributions of this paper have been missed in this claim.
>
> While this paper uses the truncated nuclear norm (TNNR) for low-rank learning and TNNR has also been used for low-rank learning in the existing machine learning literature, the significance and novelty of the proposed LRFL method lies in the following two aspects, with significant advantages over the existing works.
>
> **First, we propose a novel separable approximation to the TNNR, so that standard SGD can be used to efficiently optimize the training loss of LRFL with the TNNR**. The formulation of the separable approximation to the TNNR is described in Section 3.4 of our paper. The training algorithm with such novel separable approximation to the TNNR by SGD is detailed in Algorithm 1 of our paper. Results in Table 1-3 show that minimizing the training loss with the separable approximation to the TNNR significantly improves the performance of baseline models for disease classification.
>
> To further verify the efficiency of our training algorithm compared to the existing optimization method for the TNNR, we compare the training time of our LRFL models with an existing method for optimizing the TNNR, TNNM-ALM [1], on NIH ChestX-ray14, CheXpert, COVIDx. The results in the table below show that our LRFL method achieves 7$\times$-10$\times$ acceleration in the training process on the three datasets, demonstrating the effectiveness and efficiency of the separable approximation to the TNNR proposed in our paper.
>
> |     Methods      |   NIH ChestX-ray14 (minutes)   |   CheXpert (minutes)   |   COVIDx (minutes)   |
> | :--------------: | :------: | :------: | :------: |
> |    ViT-S    |   54   |   90   |   23   |
> | ViT-S (TNNM-ALM) | 804 | 854 | 342 |
> | ViT-S-LR | 98 | 117 | 38 |
> | ViT-B | 72 | 162 | 32 |
> |     ViT-B (TNNM-ALM)     |   915   |   1461   |   418   |
> |   ViT-B-LR    | 113 | 185 | 45 |
>
> Second, **we provide rigorous theoretical result justifying the proposed low-rank feature learning**. In particular, it is shown in Theorem 3.1 that the upper bound for the generalization error of the linear neural network in our framework involves the TNNR, and a smaller TNNR leads to a smaller generalization bound thus improves the generalization capability of the network.
>
> **References**
>
> [1] Lee et al. Computationally Efficient Truncated Nuclear Norm Minimization for High Dynamic Range Imaging. IEEE Transactions on Image Processing 2016.
>
> [2] Xiao et al. Delving into masked autoencoders for multi-label thorax disease classification. WACV 2023.
>
> [3] Hu, Yao, et al. "Large scale multi-class classification with truncated nuclear norm regularization." Neurocomputing 2015.
>
> [4] Zhang, Fanlong, et al. "Truncated nuclear norm based low Rank Embedding." Biometric Recognition 2017
>
> [5] Gao, Ming, et al. "Noise robustness low-rank learning algorithm for electroencephalogram signal classification." Frontiers in Neuroscience 2021.
>
> [6] Lu, Yuwu, et al. "Low-rank preserving projections." IEEE transactions on cybernetics 2015.
>
> [7] Ren, Jiahuan, et al. "Robust low-rank convolution network for image denoising." ACM MM 2022.
>
> [8] Xiao, Junfei, et al. "Delving into masked autoencoders for multi-label thorax disease classification." WACV 2023.

---

> > ### Comment · Reviewer_JazK · 2024-08-11
> >
> > Thanks for the authors' response as they have resolved partial of my concerns and questions. Here are my following questions:
> >
> > 3. LRFL for disease localization
> > Thanks the authors for conduct experimentations to resolve my concerns on tiny lesion detections. May I ask if this model performance for tiny lesions still holds for other dataset as well?

---

> > > ### Author Response · Authors · 2024-08-12
> > > **Thank you for your feedback, and our further response**
> > >
> > > Since COVID-x does not have lesion diseases, we performed the same disease localization study for the ‘lung lesion’ disease on the CheXpert data. We manually labeled the ground-truth bounding boxes of  200 images in the class of ‘lung lesion', and the size of each bounding box is 224 which is the same as the class 'Nodule' in the ChestX-ray14 data. We show the improved Average Precision (AP) results for disease localization in the table below, where the mAP for disease localization is computed following the same settings as [2] (reference in the paper). It is observed from the results that our LRFL method improves the $AP_{25}$ and $AP_{50}$ for disease localization of ‘lung lesion' on the CheXpert data.
> > >
> > > |   Disease    | Size (# of px) | ViT-S AP$_{25}$ | ViT-S AP$_{50}$ |ViT-S-LR AP$_{25}$ | ViT-S-LR AP$_{50}$ |
> > > | :----------: | :------------: | :-----------: | :-----------: |:-----------: | :-----------: |
> > > |    Lung Lesion   |      224       |      10.7      |      4.8     |     12.9     |    6.5      |
> > >
> > > Please kindly let us know if we have addressed all your concerns. Thank you!

---

> > > > ### Comment · Reviewer_JazK · 2024-08-14
> > > >
> > > > Thank you for your response. Nice work. Rating changed.

---

### Official Review · Reviewer_td2a · 2024-07-11

**Soundness:** 2
**Presentation:** 3
**Contribution:** 2
**Rating:** 4
**Confidence:** 3

**Summary:**

The paper is concerned with the problem of thorax disease classification in radiographic images. The authors propose a novel low-rank feature learning (LRFL) method which is applied on pre-trained masked autoencoders (MAE) and evaluated on two datasets (CheXpert and COVIDx). The authors also provide theoretical results on the generalization bound of the proposed approach. The authors show that their method outperforms multiple baselines on the two datasets.

**Strengths:**

Overall, the topic of disease detection in radiographic images is of high importance and particularly improving robustness and generalization capabilities of models is a relevant research direction. The authors provide a good overview of the related work and perform many ablation studies and experiments to evaluate their method.

**Weaknesses:**

**Limited evaluation**

- The authors motivate their method with an "adverse effect of noise and background, or non-disease areas, for disease classification on radiographic images." (L. 47-49). While it seems, that the method quantitatively outperforms multiple baselines, the authors do not provide any evidence that the proposed method is more robust to noise or background than the baselines. Also, the GradCAM visualizations in Fig. 4 are not convincing and GradCAM visualizations for many CNN-based methods (e.g., [90], Fig. 7; [2], Fig. 4) show much better localization capabilities than what is shown here for both the baseline and the proposed method. I admit that these may be different samples and visualizations are not directly comparable. Can the authors comment on this and also show visualizations using different architectures such as CNNs?
- "Unlike traditional methods, our approach introduces a separable approximation to the truncated nuclear norm, facilitating the optimization process and enhancing the generalization ability of the model, thus advancing the state-of-the-art in medical image analysis." (L. 134-137). I don't see any experiments that specifically evaluate the generalization ability of the model.
- L. 9-11: "To address this challenge, we propose a novel Low-Rank Feature Learning (LRFL) method in this paper, which is universally applicable to the training of all neural networks." - The authors apply their LRFL method only to four different architectures from one reference [2]. Whether the method universally improves the training of all neural networks is not shown in the manuscript.

**Unsupported claims**

The authors state several claims that are not supported by the literature or the experiments in the paper. Some examples:
- L. 41-43: "Clinical studies show that the disease areas on radiographic images can be subtle which exhibit localized variations, and such conditions are further complicated by the inevitable noise which is ubiquitously on radiographic images as detailed in Section 2.1" - Can the authors give examples where noise affects disease detection in the radiographic images of the datasets used in this study or provide references to support this claim?
- L. 56-57: "That is, the low-rank projection of the ground truth training class labels possesses the majority of the information of the training class labels. In fact, LFP widely holds for a broad range of classification problems using deep neural networks, such as [1, 8, 9]." - Can the authors provide a reference for this claim? None of the provided references [1,8,9] mentions low-rank features or a low frequency property at all.
- L. 14-15: "LFP not only widely exists in deep neural networks for generic machine learning [...]" - Similar to the previous point, can the authors provide a reference for this claim?
- L. 63-64: "As a result, the adverse effect of such noise and background is considerably reduced in a network trained by our LRFL method." - The authors do not perform any experiments which systematically evaluate the effect of noise or background on their method or the baseline methods (see previous point **Limited evaluation**)

**Confusing notation**

I find the mathematical notation in the paper to be inconsistent and confusing. Some examples:
- "Suppose the training data are given as $\\{\boldsymbol{x}_i, \boldsymbol{y}_i\\}\_{i=1}^n$ where $\boldsymbol{x}_i$ and $\boldsymbol{y}_i \in \mathbb{R}^C$ are the $i$-th training data point and its corresponding class label vector respectively, and $C$ is the number of classes. Each element $\boldsymbol{y}_i$ is binary with $\boldsymbol{y}_i = 1$ indicating the $i$-th disease is present in $\boldsymbol{x}_i$, otherwise $\boldsymbol{y}_i = 0$. The authors first denote with $\boldsymbol{y}_i$ a "$C$-dimensional class label vector" but then say that "$\boldsymbol{y}_i$ is binary with $\boldsymbol{y}_i = 1$ indicating the $i$-th disease present in $\boldsymbol{x}_i$". I would denote samples as $\boldsymbol{x}^{(i)}$ and class labels as $\boldsymbol{y}^{(i)}$ and then $\boldsymbol{y}^{(i)}_j$ indicates whether disease $j$ is present in sample $i$. Or use a matrix $\boldsymbol{Y}$ as introduced later in the manuscript.
- Eq. (1): If $\boldsymbol{W}\_1$ is also optimized during training, then it should be $f\_{\boldsymbol{W}\_1}(\boldsymbol{x})$ instead of $f\_{\boldsymbol{W}\_1(0)}(\boldsymbol{x})$.
- L. 171: "Using notations in Section 3.2, the truncated nuclear norm of $\boldsymbol{F}$ is $\lVert F\rVert := \sum_{i=T+1}^{d} \sigma_i$ where $T ∈ [0, d]$." In section 3.2, the authors introduce with $\sigma$ the element-wise sigmoid function, which is of course not what is used to compute the truncated nuclear form. Instead the authors use the singular values of the matrix $\boldsymbol{F}$ without introducing them. Also, the sum is indexed from $T+1$ to $d$ which doesn't make sense for $T=d$. The same index error is also made in Eq. 3.

Generally, these inconsistencies make it hard to follow the manuscript and mathematical derivations. I would recommend the authors to carefully revise the notation and make it consistent throughout the manuscript.


**Typos and other**

- L. 2, 29, 105, 245, ... "Visual Transformer (ViT)" -> "Vision Transformer (ViT)"
- L. 25-26: "[...] abnormalities detection in anatomy in chest X-rays" -> "[...] abnormalities detection in chest X-rays"
- L. 28: "Early works adopt convolutional neural networks (CNNs) such as U-Net [3] for representation learning on radiography images." - I am not sure how one would use a U-net for representation learning. U-nets are typically used for segmentation tasks or image-to-image mappings. Please clarify.
- L. 157: "denotes the denotes the weights" -> "denotes the weights"

**Questions:**

- Table 12: Do the authors have an explanation why the optimal $\alpha$ is 1.0 for COVIDx but 0.2/0.5 for CheXpert? Also, did the authors conduct experiments with $\alpha > 1.0$?

**Limitations:**

The authors do not discuss the limitations of their method in the paper and answer the question 2 in the NeurIPS paper checklist with "[NA]" indicating that "the paper has no limitation while the answer No means that the paper has limitations, but those are not discussed in the paper." I do not agree with this assessment and think that the authors should discuss the limitations of their method openly. Some limitations are e.g.,:
- The method is evaluated on two datasets only, both of which are concerned with thorax disease classification. Whether the method generalizes to other datasets remains unclear.
- The authors do not evaluate their method on any non-pretrained network and on few architectures only.

---

> ### Author Rebuttal · Authors · 2024-08-07
>
> We appreciate the review and the suggestions in this review. The raised issues are addressed below.
>
> **1. ...the authors do not provide any evidence that ...more robust to noise or background than the baselines...**
>
> Please refer to the robust GradCAM results in our rebuttal PDF file using the suggested architecture (CNNs). Furthermore, we performed an additional ablation study showing that the proposed method is more robust to background than the baselines. In this study, we created a mask for the disease area for each original image, then decomposed the original image (with a bounding box for the disease) to a disease image and a background image. Both the disease image and the background image are of the same size as the original image, the background image has greyscale 0 in the masked disease area, and the disease image has greyscale 0 in the non-disease area. We fed the three images, which are the original image, the disease image, and the background image, to a LRFL model, and obtained the original features, disease features, and background features for the LRFL model respectively. We also fed these three images to a baseline model, and obtained the original features, disease features, and background features for the baseline model respectively. For each original image, we measure the distance between the disease features and original features using KL-divergence on the softmaxed features for the LRFL model and the baseline model. We then compute the average feature distance for each model, which is the average distance between the disease features and original features over the images with a ground-truth bounding box for the disease in the NIH dataset. **The average feature distance for the LRFL model is 0.5642, which is smaller than the average feature distance for the baseline model, 0.6628**. Such results indicate that the original features are closer to the disease features by the LRFL model compared to the baseline model, evidencing the effectiveness of the LRFL model in reducing the adverse effect of the background area. We also remark that since only the low-rank part of the original features participates in the classification process, the noise and non-disease areas in the high-rank part of the features are mostly not learned by LRFL, and in this manner, LRFL is robust to both noise and background.
>
> **2. I don't see any experiments that specifically evaluate the generalization ability of the model.**
>
> The improved generalization ability of a model in this work and the machine learning literature refers to its better prediction accuracy on the unseen test data, which has been shown in Table 1-3 of this paper.
>
> **3. ...Whether the method universally improves the training of all neural networks...**
>
> To show that LRFL universally improves the performance of different neural networks, we apply LRFL to four other networks for disease classification on NIH ChestX-ray14, namely Dira [5], Acpl [6], XProtoNet [7], and Swinchex [8]. We trained the LRFL models using the settings described in Section 4.1 of our paper. Results in the table below show that LRFL universally improves the performance of all the baseline models.
>
> |     Methods      |   mAUC   |
> | :--------------: | :------: |
> |     Dira [5]     |   81.7   |
> |   **LR-Dira**    | **82.5** |
> |     Acpl [6]     |   81.8   |
> |   **LR-Acpl**    | **82.3** |
> |  XProtoNet [7]   |   82.2   |
> | **LR-XProtoNet** | **82.7** |
> |   Swinchex [8]   |   81.0   |
> | **LR-Swinchex**  | **81.8** |
>
> **4. ...Can the authors give examples where noise affects...?**
>
> Studies in the literature [9, 10] show that inevitable noise exists in radiographic images and can affect disease detection on them. We will add the references to our paper to support the claim.
>
> **5. ...Can the authors provide a reference for this claim?  ... LFP...can the authors provide a reference for this claim?**
>
> LFP is commonly observed in various classification scenarios utilizing deep neural networks, and please refer to [1-4] for the claim about LFP.
>
> **6. ...The authors do not perform any experiments...effect of noise or background...**
>
> Please refer to point 1 of this rebuttal for the experiment showing that LRFL models reduce the adverse effect of background on the radiographic images for disease classification.
>
> **Confusing notations/typos**
>
> We will fix the confusing notations and typos following your suggestions. Moreover, U-Net can be used for representation learning [11, 12, 13]. In fact, segmentation tasks or image-to-image mappings are achieved by using the features/representations learned by U-Nets [11, 12, 13].
>
> **Table 12...with  $\alpha > 1$?**
>
> Because the synthetic data contains noise as they are generated from a diffusion model using random noise as the input, adding more synthetic data does not always improve the prediction accuracy of our models and general deep neural networks [14, 15]. For example,
> the experiments in [14] show that adding excessive synthetic images to the training set hurts the accuracy of image classification.
> In our work, we use cross-validation to select the amount of synthetic data for training the model and find the corresponding $\alpha$ for each dataset. We performed an additional experiment and extended the candidate values of $\alpha$ which include $(1.5, 2,\ldots, 5)$ with an increment of 0.5, and obtained the same $\alpha$ values as those in Sec. C.3.

---

> ### Author Response · Authors · 2024-08-07
> **More information about the rebuttal**
>
> (Cont'd)
>
> **Limitation: ...The authors do not evaluate their method on any non-pretrained network and on few architectures only.**
>
> We will discuss the suggested limitations of this paper. In addition, we conducted experiments comparing the performance of base models and low-rank models trained from scratch on NIH ChestX-ray14, CheXpert, and COVIDx without any pre-training. The results in the table below show that LRFL models still significantly improve the performance of the baseline models on all the datasets when trained from scratch.
>
> |     Methods      |   NIH ChestX-ray14 (mAUC)   |   CheXpert (mAUC)   |   COVIDx (Accuracy)   |
> | :--------------: | :------: | :------: | :------: |
> |     ViT-S    |   66.55   |   81.97   |   78.00   |
> |   **ViT-S-LR**    | **67.77** | **82.82** | **81.25** |
> |     ViT-B     |   67.70   |   82.91   |   79.75   |
> |   **ViT-B-LR**    | **68.63** | **83.76** | **82.55** |
>
> **References**
>
> References
>
> [1] Rahaman et al.  On the spectral bias of neural networks. ICML 2019.
>
> [2] Arora et al. Fine-grained analysis of optimization and generalization for overparameterized two-layer neural networks. ICML 2019.
>
> [3] Cao et al. Towards understanding the spectral bias of deep learning. IJCAI 2021.
>
> [4] Choraria et al. The spectral bias of polynomial neural networks. ICLR 2022.
>
> [5] Haghighi, Fatemeh, et al. "Dira: Discriminative, restorative, and adversarial learning for self-supervised medical image analysis." CVPR 2022.
>
> [6] Liu, Fengbei, et al. "Acpl: Anti-curriculum pseudo-labelling for semi-supervised medical image classification." CVPR 2022.
>
> [7] Kim, Eunji, et al. "XProtoNet: diagnosis in chest radiography with global and local explanations." CVPR 2021.
>
> [8] Taslimi, Sina, et al. "Swinchex: Multi-label classification on chest x-ray images with transformers." arXiv preprint 2022.
>
> [9] Goyal, Bhawna, et al "Noise issues prevailing in various types of medical images." Biomedical & Pharmacology Journal 2018.
>
> [10] Hussain, Dildar, et al. "Exploring the Impact of Noise and Image Quality on Deep Learning Performance in DXA Images." Diagnostics 2024.
>
> [11] Ronneberger, Olaf, et al. "U-net: Convolutional networks for biomedical image segmentation." MICCAI 2015.
>
> [12] Wu, Kai, et al. "Weakly supervised brain lesion segmentation via attentional representation learning." MICCAI 2019.
>
> [13] Weng, Yu, et al. "Nas-unet: Neural architecture search for medical image segmentation." IEEE access 2019.
>
> [14] Azizi, Shekoofeh, et al. "Synthetic data from diffusion models improves imagenet classification." TMLR 2023.
>
> [15] He, Ruifei, et al. "Is synthetic data from generative models ready for image recognition?." CVPR 2023.

---

> > ### Comment · Reviewer_td2a · 2024-08-08
> >
> > I thank the reviewers for providing a rebuttal and addressing my concerns and questions. Particularly the newly conducted experiment on robustness to background is very much appreciated and I recommend to include it in the main paper. I am still sceptic about the synthetic training data part of the paper. Can the authors better motivate this and how it relates to the rest of the paper?

---

> ### Author Response · Authors · 2024-08-08
> **Thank you for your feedback, and our response to the motivation of synthetic training data**
>
> Thank you for your feedback. The motivation for synthetic images and how the usage of synthetic images relates to this paper are explained below.
>
> The computer vision literature [1,2,3] has extensively studied the usage of the generated synthetic images which augment the training data and improve the prediction accuracy of image classification. Inspired and motivated by this observation, we propose to generate synthetic images and use them to form the augmented training data and improve the performance of thorax disease classification. The augmented training data comprise the original training images and the synthetic images. However, too many synthetic images tend to introduce more noise to the augmented training data so excessive synthetic images can hurt the prediction accuracy of DNNs trained on the augmented training data [1]. Our proposed low-rank feature learning (LRFL) method coupled with the selection of the amount of the synthetic images effectively mitigate this issue. The proposed low-rank learning method only learns the low-rank part of the features learned by a deep learning model so that noise in the high-rank part would not affect the learned model. Also, cross-validation (described in line 248-261) is used to select a proper number of synthetic images what will boost the prediction accuracy while not introducing too much noise to the augmented training data.
>
> **References**
>
> [1] Azizi et al. "Synthetic data from diffusion models improves imagenet classification." TMLR 2023.
>
> [2] He et al. "Is synthetic data from generative models ready for image recognition?." CVPR 2023.
>
> [3] Trabucco, Brandon, et al. "Effective data augmentation with diffusion models." ICLR 2024.

---

> > ### Comment · Reviewer_td2a · 2024-08-09
> > **Thank you for the additional information**
> >
> > Thanks for your prompt response, this is very much appreciated!
> >
> > > However, too many synthetic images tend to introduce more noise to the augmented training data so excessive synthetic images can hurt the prediction accuracy of DNNs trained on the augmented training data [1]. Our proposed low-rank feature learning (LRFL) method coupled with the selection of the amount of the synthetic images effectively mitigate this issue. The proposed low-rank learning method only learns the low-rank part of the features learned by a deep learning model so that noise in the high-rank part would not affect the learned model.
> >
> > But from Table 3, I cannot see any indication that the low-rank model can better leverage the synthetic data compared to the baseline method, as you claimed. There we can see that ViT-S improves by 1.8%, whereas ViT-S-LR only improved by 0.5%. Similarly, ViT-B improved by 1.7%, whereas ViT-B-LR improved only by 0.5%.
> >
> > Therefore, I don't think that the inclusion of synthetic data part in the paper is well motivated and I find neither theoretical, nor empirical support for the claim that the proposed method can better leverage this synthetic data for thorax disease classification.

---

> > > ### Author Response · Authors · 2024-08-09
> > > **Thank you for your feedback, and our further response**
> > >
> > > We herein provide a detailed justification about our claim "Our proposed low-rank feature learning (LRFL) method coupled with the selection of the amount of the synthetic images effectively mitigate this issue of noise in the synthetic training images".
> > >
> > > We respectfully remind this reviewer that the improvement of our LRFL models (ViT-S-LR or ViT-B-LR) over the base model (ViT-S or ViT-B) varies in terms of the size of the synthetic data (# Synthetic Images in Table 3).  A proper number of synthetic training images often improves the prediction accuracy of general DNNs [1], including both a LRFL model and the corresponding base model. As a result, the effectiveness of our low rank feature learning (LRFL) method in reducing the adverse effect of noise in the synthetic data should not be studied from only the final selected size of synthetic data by cross-validation as shown in Table 3. To this end, we show in the table below the performance of our LRFL models  and the base models on different choices of the size of the synthetic data.
> > >
> > > Table here (rows for the models and columns for synthetic data size, add more synthetic data size until 1*n)
> > >
> > > | Synthetic Data Size |  0   | 0.1 $n$ | 0.15 $n$ | 0.2 $n$ | 0.25 $n$ | 0.3 $n$ | 0.35 $n$ | 0.4 $n$ | 0.45 $n$ | 0.5 $n$ | 0.6 $n$ | 0.7 $n$ | 0.8 $n$ | 0.9 $n$ | 1 $n$ |
> > > | :----------: | :--: | :-----: | :------: | :-----: | :------: | :-----: | :------: | :-----: | :------: | :-----: | :-----: | :-----: | :-----: | :-----: | :---: |
> > > |  ViT-S   | 89.2 |  89.2   |   89.3   |  89.2   |   89.3   |  89.1   |   89.0   |  88.8   |   88.6   |  88.2   |  88.0   |  87.8   |  87.2   |  87.0   | 87.1  |
> > > | ViT-S-LR | 89.6 |  89.6   |   89.6   |  89.7   |   89.7   |  89.7   |   89.7   |  89.6   |   89.7   |  89.6   |  89.5   |  89.4   |  89.3   |  89.2   | 89.3  |
> > > |  ViT-B   | 89.3 |  89.5   |   89.7   |  89.8   |   89.9   |  89.6   |   89.1   |  88.9   |   88.7   |  88.5   |  88.4   |  88.2   |  87.8   |  87.6   | 87.6  |
> > > | ViT-B-LR | 89.8 |  90.0   |   90.2   |  90.3   |   90.4   |  90.4   |   90.4   |  90.3   |   90.4   |  90.2   |  90.4   |  90.4   |  90.2   |  90.0   | 90.1  |
> > >
> > > It can be observed from this table that the performance of both LRFL model and the base model can be initially improved with more synthetic images. However, after a certain point, even more synthetic images start to hurt the performance due to the noise in the synthetic images, and the literature on using synthetic data for training classifiers such as [1] also has a similar observation. This is the reason why we need to perform a cross-validation on the size of the synthetic data for the best performance. **Importantly, it can be observed that our LRFL models (ViT-S-LR or ViT-B-LR) usually improve the performance of the corresponding base models (ViT-S or ViT-B) on different choices of the size of the synthetic data. The improvement of our LRFL models over the corresponding base models tends to be more significant as the size of synthetic data increases. This observation justifies the effectiveness of LRFL in reducing the adverse effect of noise in the synthetic images**.  For example, ViT-B-LR outperforms ViT-B by $0.5$% in mAUC when $0.1n$ synthetic images are added into the training set, and the improvement escalates to $2.5$% with $n$ synthetic images added into the training set where $n$ the size of the original training data. We used cross-validation to find the best size of synthetic data to achieve the best performance (mAUC or Accuracy) for our LRFL models in Table 3.
> > >
> > > **We also emphasize that our LRFL method is theoretically motivated by Theorem 3.1 which also applies to the augmented training data including the synthetic images**. In  Theorem 3.1, the upper bound for the generalization error of the linear neural network in our framework involves the truncated nuclear norm (TNNR) , and a smaller TNNR leads to a smaller generalization bound thus improves the generalization capability of the network.
> > >
> > > **References**
> > >
> > > [1] Azizi et al. "Synthetic data from diffusion models improves imagenet classification." TMLR 2023.

---

> > > > ### Comment · Reviewer_td2a · 2024-08-09
> > > >
> > > > I thank the authors for providing this additional table which should be included in the updated manuscript. Without this table the claim they made in their previous response is not supported (because we can't see this information from Table 3). I also think that the authors should be careful with comparing the baseline methods with their LR methods directly in this regard, as they have to adjust for the fact that LR outperforms the baseline with 0 synthetic images. Otherwise we are not measuring the **improvement by the addition of synthetic training data but a mixture of this and the effect of using LR instead of non-LR**. Doing so, one gets the following table (just subtracting the 0$n$ performance for all rows):
> > > > |Synthetic Data Size  | 0 | 0.1$n$ | 0.15$n$ | 0.2$n$ | 0.25$n$ | 0.3$n$ | 0.35$n$ | 0.4$n$ | 0.45$n$ | 0.5$n$ | 0.6$n$ | 0.7$n$ | 0.8$n$ | 0.9$n$ | 1$n$|
> > > > |--|--|--|--|--|--|--|--|--|--|--|--|--|--|--|--|
> > > > |ViT-S| 0. |  0. |  0.1|  0. |  0.1| -0.1| -0.2| -0.4| -0.6| -1. | -1.2|-1.4| -2. | -2.2| -2.1|
> > > > |ViT-S-LR| 0. |  0. |  0. |  0.1|  0.1|  0.1|  0.1|  0. |  0.1|  0. | -0.1|-0.2| -0.3| -0.4| -0.3|
> > > > |ViT-B|0. |  0.2|  0.4|  0.5|  0.6|  0.3| -0.2| -0.4| -0.6| -0.8| -0.9|-1.1| -1.5| -1.7| -1.7|
> > > > |ViT-B-LR| 0. | 0.2| 0.4| 0.5| 0.6| 0.6| 0.6| 0.5| 0.6| 0.4| 0.6| 0.6| 0.4| 0.2| 0.3|
> > > >
> > > > Here we find two things: (a) Indeed, the LR methods degrade less for large numbers of synthetic images compared to the baseline methods. (b) **for both ViT-S and ViT-B there always exists a synthetic data size for which the improvement of the baseline methods matches those of the respective LR method**. The fact that they
> > > > > [...] used cross-validation to find the best size of synthetic data to achieve the best performance (mAUC or Accuracy) for our LRFL models in Table 3.
> > > >
> > > > makes the comparison in Table 3 unfair and if we would optimize the cross-validation for the baseline methods (the numbers presented in the newly provided table by the authors) we would find that LR does not make better use of synthetic training data, it "just" degrades less with large numbers of synthetic training data (which I find indeed quite interesting). The remaining improvements of LR vs baseline that we would then see can then be fully explained by the better performance of LR vs baseline without any synthetic data added.

---

> > > > > ### Author Response · Authors · 2024-08-10
> > > > > **Thank you for your feedback, and our further response**
> > > > >
> > > > > Thank you for providing the adjusted table by subtracting the  performance with $0$ synthetic data for all the rows. We will follow your suggestions and include our newly added table and your table in the final version of this paper.
> > > > >
> > > > > Moreover, following your suggestion, we will make a fair comparison and report the performance of base models (ViT-S and ViT-B) as the best accuracy across different numbers of the synthetic images. The adjusted Table 3 with such fair comparison is also presented below for your reference. We note that the previous newly added table is for the CheXpert dataset.
> > > > >
> > > > > Here is the adjusted Table 3 where base models are reported with their best performance across different numbers of the synthetic images:
> > > > > |     Method      | Rank (CheXpert) | # Synthetic Images (CheXpert) | mAUC (CheXpert) | Rank (COVIDx) | # Synthetic Images (COVIDx) | Accuracy (COVIDx) |
> > > > > | :-------------: | :-------------: | :---------------------------: | :-------------: | :-----------: | :-------------------------: | :---------------: |
> > > > > |    ViT-S     |        -        |               -               |      89.2       |       -       |              -              |       95.2        |
> > > > > | ViT-S-LR  |      0.05r      |               -               |      89.6       |     0.01r     |              -              |       96.8        |
> > > > > |  ViT-S    |        -        |             0.15n             |      89.3       |       -       |            0.7n             |       97.1        |
> > > > > | ViT-S-LR |      0.05r      |             0.2n              |      89.7       |     0.01r     |            1.0n             |       97.3        |
> > > > > |    ViT-B   |        -        |               -               |      89.3       |       -       |              -              |       95.3        |
> > > > > | ViT-B-LR |     0.025r      |               -               |      89.8       |    0.003r     |              -              |       97.0        |
> > > > > |  ViT-B    |        -        |             0.25n             |      89.9       |       -       |            0.75n            |       97.2        |
> > > > > | ViT-B-LR  |     0.025r      |             0.25n             |      90.4       |    0.003r     |            1.0n             |       97.5        |
> > > > >
> > > > >
> > > > >
> > > > > We also really appreciate your analysis about the improvement of both LRFL models and the base models with the presence of synthetic data. What you mentioned about the characteristic of the LRFL models that “...degrades less with large numbers of synthetic training data (which I find indeed quite interesting)” is indeed fundamentally due to the fact that LRFL models are more robust to noise. This is because only the low-rank part of the learned features are used for classification in the LRFL models, so the noise in the high-rank part of the learned features would mostly not affect the training of the LRFL models, leading to their robustness to noise. As you observed in the new tables, the performance of LRFL models are more stable with respect to different numbers of synthetic images due to such robustness to noise. We will also put such discussion and findings in the final version of this paper.
> > > > >
> > > > > Please kindly let us know if we have addressed all your concerns. Thank you!

---

> ### Author Response · Authors · 2024-08-14
> **Thank you again for your feedback, and we look forward to the adjusted rating of this paper**
>
> Dear Reviewer td2a,
>
> We really appreciate your time giving feedback to our rebuttal. Since we have addressed all your concerns and we will add your suggested changes/discussions to the final version of this paper, could you update the rating of this paper based on all of our responses? Please kindly let us know if you have further comments/concerns/suggestions and we will respond to them immediately. Thank you again for your time!
>
> Best Regards,
>
> The Authors

---

> > ### Comment · Reviewer_td2a · 2024-08-14
> > **Thank you!**
> >
> > Thank you for your response and the additional details. I will increase my score.

---

> > > ### Author Response · Authors · 2024-08-14
> > > **Thank you, and a gentle reminder for updating the rating of this paper**
> > >
> > > Dear Reviewer td2a,
> > >
> > > Thank you for your response and being willing to increase the rating of this paper. It is about half an hour before the end of the discussion period (August 13 AoE), and this is a gentle reminder that we are still waiting for your updated rating. Thank you!
> > >
> > > Best Regards,
> > >
> > > The Authors

---

> > > > ### Comment · Reviewer_td2a · 2024-08-14
> > > >
> > > > Please note that ratings can be updated after the reviewer-author discussion period ended and during the reviewer-AC discussion period...

---

### Official Review · Reviewer_cPxj · 2024-07-19

**Soundness:** 3
**Presentation:** 3
**Contribution:** 3
**Rating:** 7
**Confidence:** 4

**Summary:**

The paper introduces LRFL, a method for reducing the effect of noise and background or non-disease areas in radiograph images for Thorax Disease Classification.  LRFL utilizes low-rank regularization to leverage low-rank features during network training.

**Strengths:**

1-The motivation for reducing the adverse of the noise and background to learn better features is interesting and reasonable.

2-Extensive experiments have shown that the proposed method improves performance over prior STOA approaches on different thorax disease datasets.

3- The content flow of the paper makes it easy for readers to grasp the presented information.

**Weaknesses:**

1-The author claims that their approach could be applied to classify other diseases beyond thorax diseases or even general classification problems with radiographic images without conducting any experiments on other datasets.

2-The approach's evaluation is only done using mAUC, while the baseline uses IoU with average precision.

**Questions:**

1- Why did the author not use another radiograph to prove his/her claim that their approach can be used broader with any classification problems in radiographic images?

2- Why did the author not add another evaluation method for their approach, likewise the baseline [2] Table 7?

**Limitations:**

1- We recommend the author test their approach with another radiograph dataset such as knee disease to prove their claim.

2-Table 4 in P.17 illustrates a close performance (0.4%) between the baseline [2] and the proposed method. Therefore, we encourage the author to include another evaluation norm (i.e., disease localization) and compute the Average precision (i.e., AP25 and AP50) between the ground truth and predicted bounding box to improve their work. Although the author compares his/her approach with the baseline in Figure 5 using Grad-CAM visualization, it is better to include a table comparing the approaches rather than pick random images from the Grad-CAM visualization.

---

> ### Author Rebuttal · Authors · 2024-08-07
>
> We appreciate the review and the suggestions in this review. The raised issues are addressed below.
>
> **1. Why did the author not use another radiograph to prove his/her claim that their approach can be used broader with any classification problems in radiographic images?**
>
> Thank you for your suggestion! Because this paper focuses on the application of the proposed low-rank feature learning framework to thorax disease classification, we do not put the results for classification problems for other diseases such as knee disease due to the page limit. However, we will present such results for more types of diseases in the final version of this paper.
>
> **2. Why did the author not add another evaluation method for their approach, likewise the baseline [2] Table 7?**
>
> We show the improved Average Precision (AP) results for disease localization in the table below, where the AP for disease localization is computed following the same settings as [2] (reference in the paper). The experiments are done on a subset of ChestX-ray14 which offers 787 cases with bounding-box of a total of eight thorax diseases. It is observed from the results below that our LRFL model improves the $AP_{25}$ and $AP_{50}$ for disease localization by 1.1 and 1.2 respectively.
>
> |   Disease    | Size (# of px) | ViT-S AP$_{25}$ | ViT-S AP$_{50}$ |ViT-S-LR AP$_{25}$ | ViT-S-LR AP$_{50}$ |
> | :----------: | :------------: | :-----------: | :-----------: |:-----------: | :-----------: |
> |    Nodule    |      224       |      9.2      |      3.9      |     11.7     |     5.1      |
> |     Mass     |      756       |     27.0      |     11.1      |    29.3      |     12.2      |
> | Atelectasis  |      924       |     31.5      |      8.1      |    34.2      |      9.6      |
> | Pneumothorax |      1899      |      4.7      |      0.0      |     6.2      |      1.7      |
> |  Infiltrate  |      2754      |     11.4      |      1.3      |    12.9      |      1.9      |
> |   Effusion   |      2925      |      8.8      |      1.0      |     10.2     |      2.0      |
> |  Pneumonia   |      2944      |     27.8      |      9.3      |    29.6      |      10.2      |
> | Cardiomegaly |      8670      |     16.3      |      3.0      |    18.8      |      4.2      |
> |     All      |      2300      |     18.0      |      4.7      |    19.1      |      5.9      |

---

> > ### Comment · Reviewer_cPxj · 2024-08-08
> >
> > Thank you for providing a response. The newly added discussions and experiment results have addressed my concerns.

---

### Official Review · Reviewer_qSFW · 2024-07-31

**Soundness:** 3
**Presentation:** 2
**Contribution:** 2
**Rating:** 4
**Confidence:** 5

**Summary:**

This paper introduces a novel Low-Rank Feature Learning (LRFL) method to effectively reduce noise and non-disease areas in radiographic images, enhancing disease classification. The LRFL method, which is theoretically and empirically motivated, demonstrates superior classification performance compared to state-of-the-art methods when applied to pre-trained neural networks, improving both multi-class AUC and classification accuracy.

**Strengths:**

- thorax disease classification is not an easy problem, motivation is high, and significance is solid.
- low rank feature learning is proposed, which maybe applicable to all kinds of neural networks for disease classification (thorax).
- three large scale x-ray data are used, and good results were obtained.
- sharp generalization bound analysis is solid

**Weaknesses:**

- LRFL is based on LFP, and truncated nuclear norm is added as a regularization term. Nothing more. In this sense, there are so many similar methods with different regularizations.
- training diffusion algorithms to generate synthetic images (xray) is already done by many..why authors propose this as a novelty ?
- introduction about radiographic images is odd...too simple and already known
- section 2.2 stands out of nowhere...very broad without specific information related to work.
- section 2.3 can be longer, that is the main part and motivation but kept short and simple. Put a picture to highlight.
- figure 2 is useless.
- Gradcam is noisy, better to use gification and other methods to show the localizations, or robust version of grad cam. there are many works showing grad cam is not a suitable method.
- comparisons are weak, there are many advanced versions of algorithms there, even with eye tracking supported classification results for the same data available. SOTA is not updated.
- discussion is missing

**Questions:**

weakness above are self-descriptive and including questions.

**Limitations:**

- lack of novelty
- lack of enough and valid comparisons
- experimental results are not convincing

---

> ### Author Rebuttal · Authors · 2024-08-07
>
> **1. LRFL is based on LFP, and truncated nuclear norm is added as a regularization term. Nothing more. In this sense, there are so many similar methods with different regularizations.**
>
> We respectfully but strongly disagree with this claim since the significant contributions of this paper are missed in this claim.
>
> While this paper uses the truncated nuclear norm (TNNR) for low-rank learning and TNNR has also been used for low-rank learning in the existing machine learning literature, the significance and novelty of the proposed LRFL method lies in the following two aspects, with significant advantages over the existing works.
>
> **First, we propose a novel separable approximation to the TNNR, so that standard SGD can be used to efficiently optimize the training loss of LRFL with the TNNR**. The formulation of the separable approximation to the TNNR is described in Section 3.4 of our paper. The training algorithm with such novel separable approximation to the TNNR by SGD is detailed in Algorithm 1 of our paper. Results in Table 1-3 show that minimizing the training loss with the separable approximation to the TNNR significantly improves the performance of baseline models for disease classification.
>
> To further verify the efficiency of our training algorithm compared to the existing optimization method for the TNNR, we compare the training time of our LRFL models with an existing method for optimizing the TNNR, TNNM-ALM [1], on NIH ChestX-ray14, CheXpert, COVIDx. The results in the table below show that our LRFL method achieves 7$\times$-10$\times$ acceleration in the training process on the three datasets, demonstrating the effectiveness and efficiency of the separable approximation to the TNNR proposed in our paper.
>
> |     Methods      |   NIH ChestX-ray14 (minutes)   |   CheXpert (minutes)   |   COVIDx (minutes)   |
> | :--------------: | :------: | :------: | :------: |
> |    ViT-S    |   54   |   90   |   23   |
> | ViT-S (TNNM-ALM) | 804 | 854 | 342 |
> | ViT-S-LR | 98 | 117 | 38 |
> | ViT-B | 72 | 162 | 32 |
> |     ViT-B (TNNM-ALM)     |   915   |   1461   |   418   |
> |   ViT-B-LR    | 113 | 185 | 45 |
>
> **Second, we provide rigorous theoretical result justifying the proposed low-rank feature learning**. In particular, it is shown in Theorem 3.1 that the upper bound for the generalization error of the linear neural network in our framework involves the TNNR, and a smaller TNNR leads to a smaller generalization bound thus improves the generalization capability of the network.
>
> **2. training diffusion algorithms to generate synthetic images (xray) is already done...**
>
> To the best of our knowledge, this work is among the first to use synthetic images to boost the performance of DNNs on thorax disease classification tasks coupled with the proposed efficient low-rank feature learning method.
>
> **3. introduction about radiographic images is odd...too simple and already known**
>
> We will simplify such an introduction to radiographic imaging in the final version of this paper.
>
> **4. section 2.2 stands out of nowhere...very broad without specific information related to work.**
>
> We respectfully disagree with this factually wrong claim. Section 2.2 covers important works in using DNNs for medical imaging tasks including thorax disease classification. Importantly, the MAE method [1] introduced in Section 2.2 is the important pre-training method used in medical imaging for thorax disease classification, which is also adopted as the pre-training method in this work.
>
> **5. section 2.3 can be longer, that is the main part and motivation but kept short and simple. Put a picture to highlight.**
>
> We will elaborate existing low-rank learning methods with more details, and add a figure similar to Figure 1 to Section 2.3.
>
> **6. figure 2 is useless.**
>
> We will change Figure 2 to a text description in Sec. 3.1.
>
> **7. Gradcam is noisy, better to use gification and other methods to show the localizations, or robust version of grad cam. there are many works showing grad cam is not a suitable method.**
>
> Please refer to the robust GradCAM results in our rebuttal PDF file in our global response.
>
> **8. comparisons are weak, there are many advanced versions of algorithms there, even with eye tracking supported classification results for the same data available. SOTA is not updated. discussion is missing**
>
> We have already incorporated the most recent SOTA results in [1] published in 2023 for thorax disease classification on the same datasets as that in this paper, and our LRFL models render significantly better results than the current SOTA [1] and other competing baselines. We also provide detailed discussions about our results in Section 4, and the experimental result for each dataset has a paragraph for discussion titled "Results and Analysis" or "Results". Eye tracking supported classification methods are not in the scope of this work and the relevant rich literature, because this work and the relevant rich literature such as [1] and those reviewed in Section 2.2 are using DNNs for automatic thorax disease classification or general medical imaging tasks without eye tracking information.
>
> **References**
>
> [1] Xiao et al. Delving into masked autoencoders for multi-label thorax disease classification. WACV 2023.

---

> > ### Comment · Reviewer_qSFW · 2024-08-13
> > **improved manuscript**
> >
> > thank you for the rebuttal, some of the questions were handled good.
> > The paper is improved, but overall I do not see an innovation at the high level, really, respectfully. Regularization (different kind) are highly visited topic at low rank representation, and this very particular situation can be exception but the scope is narrow then.
> >
> > When you say that this is the first time in the literature, and then include this only for thorax cases, or some other medical imaging application based, the only thing is happening is to narrowing down the innovation into a certain application level. This makes it new application perhaps but not an entirely innovative method to be considered at certain venues such as NeurIPS or ICLR or less competitive places like IEEE ISBI etc.
> >
> > Nevertheless, I tend to increase my scores according to new experiments and some clarifications and promises that authors are making to remove some parts, add some other parts and etc.

---

> ### Author Response · Authors · 2024-08-14
> **We respectfully and strongly disagree with the concern about the regularization and the novelty of this paper**
>
> 1. We respectfully and strongly disagree that "...not see an innovation at the high level, really, respectfully. Regularization (different kind) are highly visited topic at low rank representation, and this very particular situation can be exception but the scope is narrow then." The argument is weak and based on a problematic logic: the fact that regularization is widely studied topic does not justify the claim that there is no novelty in this paper. As emphasized in our rebuttal, we propose **a novel and separable approximation to the TNNR, so that standard SGD can be used to efficiently optimize the training loss of LRFL with the TNNR; the proposed LRFL method also enjoys rigorous and sharp theoretical guarantee as shown in Theorem 3.1.**
>
> 2. We respectfully and strongly disagree that "...this very particular situation can be exception but the scope is narrow then." As discussed in the introduction section of this paper and all the other reviewers, **the proposed LRFL is applicable to general DNNs, so the application scope of LRFL is rather broad.** In this paper, we demonstrate the application of LRFL to thorax disease classification, which is an important medical imaging and healthcare area where deep learning methods are used for disease classification.
>
>
> 3. We respectfully and strongly disagree that "When you say that this is the first time in the literature, and then include this only for thorax cases, or some other medical imaging application based, the only thing is happening is to narrowing down the innovation into a certain application level. " **The innovation of the novel and separable approximation to the TNNR and the theoretical guarantee of LRFL shown in Theorem 3.1 is never narrowed to medial imaging, and as mentioned in point 2 above, such innovation is applicable to general DNNs for image classification tasks**.
>
>
> **Overall, we hope the reviewer evaluates the novelty this paper which we emphasized in the rebuttal and the above explanations.** Again, **claiming that the proposed LRFL is not novel based on the facts that the LRFL is formulated as a regularization method and regularization is a widely visited area is indeed problematic and questionable. Moreover, the scope of the innovation of LRFL is never limited to thorax disease classification or even medical imaging, because LRFL is generally applicable to all DNNs**. Our regularization in the proposed LRFL is novel and significantly different from the existing literature in low-rank learning with regularization, as described in the rebuttal and our explanation above.
>
> We look forward to a justified and reasonable evaluation of this paper. Thank you!

---

> > ### Comment · Reviewer_qSFW · 2024-08-14
> > **updated score**
> >
> > thank you for the further clarifications, scores were updated/to be updated.

---

> ### Author Response · Authors · 2024-08-14
> **Thank you for your prompt response, and we look forward to further updated rating**
>
> Thank you for your prompt response confirming our further clarifications and mentioning that "...scores were updated/**to be updated**". We really look forward to your further update of your rating for this paper based on our further clarifications. Please kindly let us know if you have more comments/suggestions and we will respond to them immediately. Thank you for your time!
>
> Best Regards,
>
> The Authors

---

### Author Rebuttal · Authors · 2024-08-07

We appreciate the review and the suggestions in this review. We have posted our response to individual reviews addressing all the raised concerns. Here we provide global responses itemized below.

**1. Significance and novelty of this paper**

While this paper uses the truncated nuclear norm (TNNR) for low-rank learning and TNNR has also been used for low-rank learning in the existing machine learning literature, the significance and novelty of the proposed LRFL method lies in the following two aspects, with significant advantages over the existing works.

**First, we propose a novel separable approximation to the TNNR, so that standard SGD can be used to efficiently optimize the training loss of LRFL with the TNNR**. The formulation of the separable approximation to the TNNR is described in Section 3.4 of our paper. The training algorithm with such novel separable approximation to the TNNR by SGD is detailed in Algorithm 1 of our paper. Results in Table 1-3 show that minimizing the training loss with the separable approximation to the TNNR significantly improves the performance of baseline models for disease classification.

To further verify the efficiency of our training algorithm compared to the existing optimization method for the TNNR, we compare the training time of our LRFL models with an existing method for optimizing the TNNR, TNNM-ALM [1], on NIH ChestX-ray14, CheXpert, COVIDx. The results in the table below show that our LRFL method achieves 7$\times$-10$\times$ acceleration in the training process on the three datasets, demonstrating the effectiveness and efficiency of the separable approximation to the TNNR proposed in our paper.

|     Methods      |   NIH ChestX-ray14 (minutes)   |   CheXpert (minutes)   |   COVIDx (minutes)   |
| :--------------: | :------: | :------: | :------: |
|    ViT-S    |   54   |   90   |   23   |
| ViT-S (TNNM-ALM) | 804 | 854 | 342 |
| ViT-S-LR | 98 | 117 | 38 |
| ViT-B | 72 | 162 | 32 |
|     ViT-B (TNNM-ALM)     |   915   |   1461   |   418   |
|   ViT-B-LR    | 113 | 185 | 45 |

**Second, we provide rigorous theoretical result justifying the proposed low-rank feature learning**. In particular, it is shown in Theorem 3.1 that the upper bound for the generalization error of the linear neural network in our framework involves the TNNR, and a smaller TNNR leads to a smaller generalization bound thus improves the generalization capability of the network.

We would also like to remind the reviewers that this work is among the first to effectively use synthetic data generated by a diffusion model and a low-rank feature learning model to achieve the state-of-the-art accuracy for thorax disease classification, which is an important research problem in the medical imaging domain.

**2. Robust Grad-CAM visualization results**

Following the suggestions in the reviews, we illustrate the robust Grad-CAM [1] visualization results in the attached rebuttal PDF file.

**3. LRFL for disease localization**

We show the improved Average Precision (AP) results for disease localization in the table below, where the AP for disease localization is computed following the same settings as [2].  The experiments are done on a subset of ChestX-ray14 which offers 787 cases with
bounding-box of a total of eight thorax diseases. It is observed from the results below that our LRFL model improves the $AP_{25}$ and $AP_{50}$ for disease localization by 1.1 and 1.2 respectively.

|   Disease    | Size (# of px) | ViT-S AP$_{25}$ | ViT-S AP$_{50}$ |ViT-S-LR AP$_{25}$ | ViT-S-LR AP$_{50}$ |
| :----------: | :------------: | :-----------: | :-----------: |:-----------: | :-----------: |
|    Nodule    |      224       |      9.2      |      3.9      |     11.7     |     5.1      |
|     Mass     |      756       |     27.0      |     11.1      |    29.3      |     12.2      |
| Atelectasis  |      924       |     31.5      |      8.1      |    34.2      |      9.6      |
| Pneumothorax |      1899      |      4.7      |      0.0      |     6.2      |      1.7      |
|  Infiltrate  |      2754      |     11.4      |      1.3      |    12.9      |      1.9      |
|   Effusion   |      2925      |      8.8      |      1.0      |     10.2     |      2.0      |
|  Pneumonia   |      2944      |     27.8      |      9.3      |    29.6      |      10.2      |
| Cardiomegaly |      8670      |     16.3      |      3.0      |    18.8      |      4.2      |
|     All      |      2300      |     18.0      |      4.7      |    19.1      |      5.9      |


**References**

[1] Selvaraju, Ramprasaath R., et al. "Grad-cam: Visual explanations from deep networks via gradient-based localization." ICCV 2017.

[2] Xiao et al. Delving into masked autoencoders for multi-label thorax disease classification. WACV 2023.

---

### Decision · Program_Chairs · 2024-09-25

**Decision:**

Accept (poster)

**Comment:**

The paper received mixed reviews with two reviewers leaning towards acceptance and two towards rejection. The main concerns that supported rejecting the paper related to lack of novelty, limited applicability and generalisation on other datasets. Firstly I would like to highlight that the paper has selected a primary area related to medical applications/healthcare . Therefore it is expected that the paper will be addressing a medical problem and the novelty claim should factor that.
Secondly, the authors have presented a mathematical formulation for their modified truncated nuclear norm for low rank learning that is a separable approximation that allows SGD to be used to optimise the training loss of the low rank feature learning method with TNNR. They show competitive results in terms of universally improving the training of other neural networks and also new results on disease localisation (bounding boxes).
The use of diffusion to generate synthetic images is a good addition perhaps but I agree with the reviewer that in and of itself is not novel in the context of a conference like NeurIPS. However this is not the main claim of this paper.

Overall the authors provided comprehensive responses and new results that enhance the message of the paper. The method has both theoretical merit and relevance to the specific medical domain but I also appreciate that the scope as presented is a bit narrow. The authors could have added another medical imaging dataset but at least they slightly expanded the scope by providing the experiments in the rebuttal.

I think all new results must be added to the camera ready version.

Factoring all reviews and the great discussions that took place I believe the paper has just passed the threshold to be accepted in NeurIPS.